# Identification and reconstitution of the rubber biosynthetic machinery on rubber particles from *Hevea brasiliensis*

Satoshi Yamashita[1†], Haruhiko Yamaguchi[2], Toshiyuki Waki[1], Yuichi Aoki[1‡], Makie Mizuno[1], Fumihiro Yanbe[1], Tomoki Ishii[1], Ayuta Funaki[1], Yuzuru Tozawa[3], Yukino Miyagi-Inoue[2], Kazuhisa Fushihara[2], Toru Nakayama[1], Seiji Takahashi[1*]

[1]Graduate School of Engineering, Tohoku University, Sendai, Japan; [2]Sumitomo Rubber Industries, Ltd, Kobe, Japan; [3]Graduate School of Science and Engineering, Saitama University, Saitama, Japan

*For correspondence: takahasi@seika.che.tohoku.ac.jp

Present address: †Graduate School of Natural Science and Technology, Kanazawa University, Kakuma, Kanazawa, Japan; ‡Graduate School of Information Sciences, Tohoku University, Sendai, Japan

Competing interests: The authors declare that no competing interests exist.

**Abstract** Natural rubber (NR) is stored in latex as rubber particles (RPs), rubber molecules surrounded by a lipid monolayer. Rubber transferase (RTase), the enzyme responsible for NR biosynthesis, is believed to be a member of the *cis*-prenyltransferase (cPT) family. However, none of the recombinant cPTs have shown RTase activity independently. We show that HRT1, a cPT from *Heveabrasiliensis*, exhibits distinct RTase activity in vitro only when it is introduced on detergent-washed *Hevea*RPs (WRPs) by a cell-free translation-coupled system. Using this system, a heterologous cPT from *Lactucasativa* also exhibited RTase activity, indicating proper introduction of cPT on RP is the key to reconstitute active RTase. RP proteomics and interaction network analyses revealed the formation of the protein complex consisting of HRT1, rubber elongation factor (REF) and HRT1-REF BRIDGING PROTEIN. The RTase activity enhancement observed for the complex assembled on WRPs indicates the HRT1-containing complex functions as the NR biosynthetic machinery.

## Introduction

Natural rubber (NR) is a non-fungible natural polymer used for manufacturing rubber products such as tires because of its unique physical properties. No other synthetic polymer has been developed to date, with physical properties comparable to NR. Although more than 2500 plant species are known to biosynthesize NR (*Metcalfe, 1967*; *van Beilen and Poirier, 2007*), industrial production of NR depends solely on latex, the cytoplasm of highly specialized cells known as laticifers harvested from the Para rubber tree *Hevea brasiliensis*. Despite its long history of use as an industrially essential natural material, the molecular mechanisms underlying NR biosynthesis, especially those required for the backbone structure formation, remain to be elucidated.

   NR exists in latex as rubber particles (RPs), which primarily consist of a hydrophobic rubber core enclosed by a lipid monolayer (*Cornish et al., 1999*). In the 1960s, the enzyme responsible for NR biosynthesis from RPs was reported to be rubber transferase (RTase) [EC.2.5.1.20] (*Archer, 1969*), although its proteinaceous components and catalytic mechanism were not elucidated. The backbone structure of NR comprises of a *cis*-1,4-polyisoprene with two or three *trans*-isoprene units at the ω-terminus (*Tanaka et al., 1996*; *Eng et al., 1994*) (*Figure 1*). Generally, the basic backbone structures of *cis,trans*-mixed polyisoprenoid alcohols, polyprenols and dolichols ($C_{30}$–$C_{120}$), are biosynthesized via sequential *cis*-condensation of the $C_5$ isoprene unit, isopentenyl diphosphate (IPP), onto all-*trans* short-chain prenyl diphosphates such as geranyl diphosphate (GPP, $C_{10}$) *E,E*-farnesyl diphosphate (FPP, $C_{15}$), and *E,E,E*-geranylgeranyl diphosphate (GGPP, $C_{20}$). This reaction is catalyzed by *cis*-

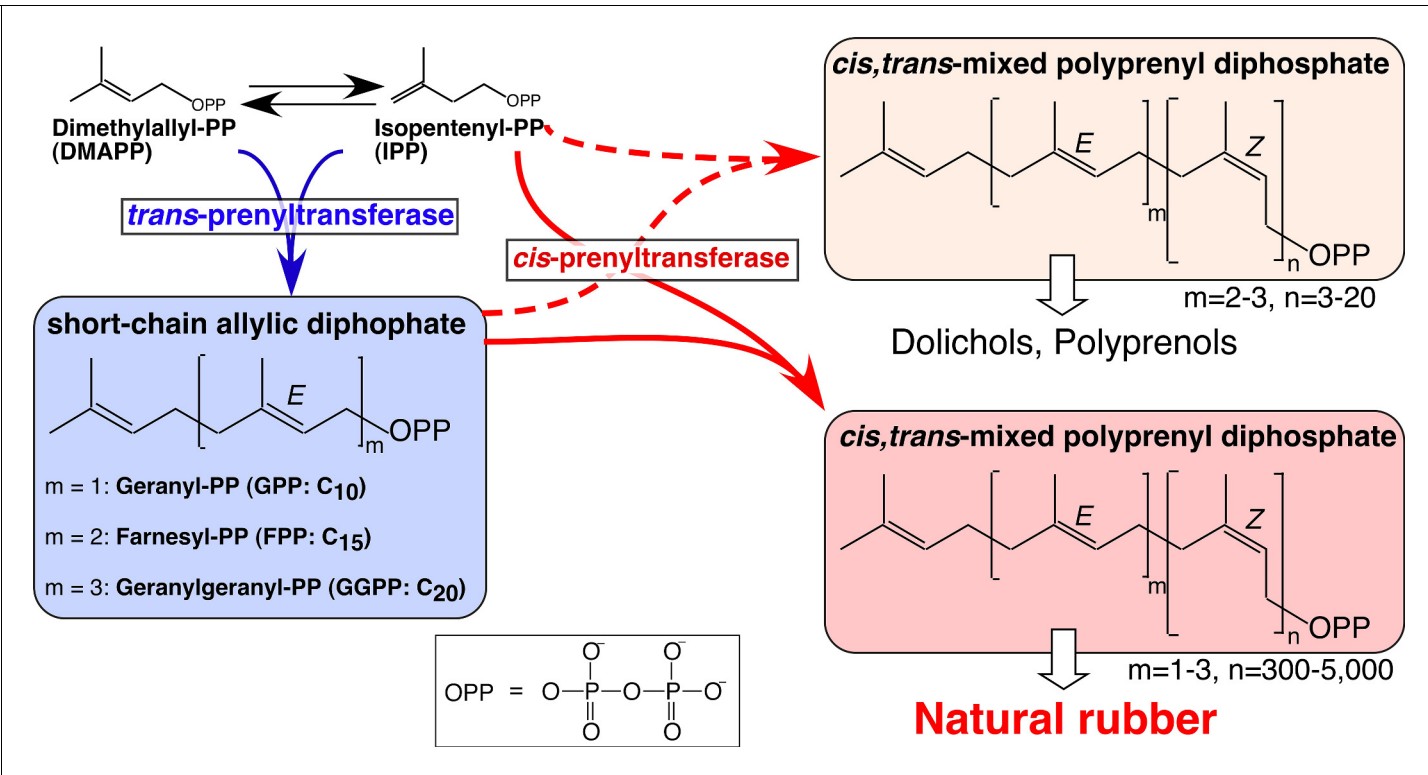

**Figure 1.** Biosynthetic pathways of backbone structures of *cis,trans*-mixed polyisoprenoids and natural rubber, catalyzed by *trans*- and *cis*-prenyltransferases.

prenyltransferase (cPT) (*Takahashi and Koyama, 2006*; *Koyama, 1999*) (**Figure 1**). The common structural unit of polyisoprenoid alcohols and NR suggests that RTase is a member of the cPT family, although the NR carbon chain length ($C_{>1500}$) is much longer than that of other *cis,trans*-mixed polyisoprenoid alcohols, such as dolichols which are indispensable as sugar carrier lipids during the protein glycosylations in the endoplasmic reticulum (ER) (*Helenius and Aebi, 2004*).

We previously isolated two cPT homologues, designated *HRT1* and *HRT2*, from *H. brasiliensis*, which are predominantly expressed in latex (*Asawatreratanakul et al., 2003*). Recombinant proteins of HRT1 and HRT2 expressed in *Escherichia coli* did not exhibit distinct cPT activity independently in vitro. While the recombinant proteins of HRT1 and HRT2, expressed in yeast and *Arabidopsis* showed significant activity, producing $C_{80-100}$ polyisoprenoids, they failed to produce longer chain products corresponding to NR (*Takahashi et al., 2012*). Recently, cPT homologues expressed in latex of other NR-producing plants, such as Russian dandelion (*Taraxacum koksaghyz*) (*Schmidt et al., 2010a*; *Schmidt et al., 2010b*), lettuce (*Lactuca sativa*) (*Qu et al., 2015*) and *Euphorbia characias* (*Spanò et al., 2015*), have been isolated, although their recombinant proteins show no RTase activity in vitro. RNA interference-mediated knock-down of cPTs expressed in the latex of *Taraxacum brevicorniculatum* have been shown to decrease latex rubber content (*Post et al., 2012*), suggesting the involvement of cPT in NR formation. However, there is no direct evidence to date suggesting that the cPTs expressed in latex are RTases. Therefore, latex-specific cPTs are thought to require eukaryotic cell-specific factor(s) to exhibit cPT activity and additional factor(s) in the latex to modify these cPTs into RTases.

With regard to activation of eukaryotic cPTs, a family of Nogo-B receptor (NgBR) performs the role of regulator of cPT activity. NgBR from humans, originally identified as a receptor for the N-terminal portion of reticulon 4B/Nogo-B (*Miao et al., 2006*), contains three predicted transmembrane domains in its N-terminal region, and a domain exhibiting low similarity to cPT in the C-terminal half, although it does not have catalytic residues conserved among cPTs. In the cPT-like domain, NgBR can also interact with a human cPT HDS/hCIT (*Shridas et al., 2003*; *Endo et al., 2003*) to stabilize it

on the ER membrane, resulting in cPT activity enhancement (*Harrison et al., 2011*). Knock-down (*Harrison et al., 2011*) or a mutation (*Eng et al., 1994*) of *NgBR* causes defects in protein *N*-glycosylation as a consequence of a reduction in the level of dolichyl phosphates and dolichol-linked oligosaccharides, indicating that NgBR is an HDS-interacting protein and this interaction is critical for its activity. *NUS1*, a yeast homolog of NgBR, is implicated to have an important role in *N*-glycosylation and cPT activity (*Park et al., 2014*). It has also been reported that a defect in *LEW1*, a counterpart of NgBR in *Arabidopsis thaliana*, results in reduced levels of dolichol contents and protein glycosylation (*Zhang et al., 2008*). Despite its importance in the eukaryotic *cis,trans*-mixed polyisoprenoid biosynthesis, a precise role of NgBR in cPT activity is still unknown owing to a lack of in vitro studies on the function of NgBR family with purified proteins. Moreover, its role as an essential subunit of a deduced cPT-containing protein complex is controversial as various eukaryotic cPTs heterologously expressed in *Saccharomyces cerevisiae* have shown distinct activity without co-expression of their partner NgBR family protein (*Asawatreratanakul et al., 2003*; *Takahashi et al., 2012*; *Schmidt et al., 2010b*; *Shridas et al., 2003*; *Endo et al., 2003*; *Harrison et al., 2011*; *Cunillera et al., 2000*; *Akhtar et al., 2013*; *Kera et al., 2012*; *Surmacz et al., 2014*).

In this study, we identified a protein complex that function as the NR biosynthetic machinery on RPs. We identified the partner proteins of HRT1 by RP proteomics and interaction network assays, revealing the formation of a protein complex consisting of HRT1 and two RP-bound proteins, where one of the two was a *Hevea* NgBR homolog. To investigate functions of the partner proteins, we developed a cell-free translation-coupled recombinant protein introduction system on detergent-washed *Hevea* RPs (WRPs). This enabled de novo synthesis of NR in vitro by the recombinant protein, providing the first direct evidence that HRT1 is an RTase in *H. brasiliensis*. In this system, a heterologous cPT from another rubber-producing plant, introduced on *Hevea* RPs, also exhibited RTase activity comparable to that of HRT1, indicating that accurate introduction of cPT on the WRPs is key to the functional expression of RTase. Reconstitution assays of HRT1 and the partner proteins on WRPs illustrated that the formation of an HRT1-containing ternary complex is important for efficient rubber production via NR biosynthesis from RPs.

## Results

### Identification of a Nogo-B receptor homologue from *H. brasiliensis* latex

Based on the hypothesis that the indispensable factor for RTase activity might be proteins on RPs, we undertook a comprehensive identification of proteins on RPs from *H. brasiliensis* latex. Before the proteomic analyses, proteins unnecessary for RTase activity were removed from RPs by washing with CHAPS as it was the best detergent for removing protein from the RPs and had a relatively low inhibitory effect on the RTase activity among the 15 detergents tested (*Figure 2—figure supplement 1*). Stepwise washing with buffers containing CHAPS resulted in washing-out a lot of the RP proteins (*Figure 2A*), whereas 80% of the RTase activity was retained on the detergent-washed RPs (*Figure 2B*). Proteome analyses were conducted after removal of the protein bands corresponding to the well-studied RP-abundant hydrophobic proteins, REF (14.6 kDa) (*Dennis and Light, 1989*) and small rubber particle protein (SRPP, 24 kDa) (*Oh et al., 1999*), to detect low-abundance proteins. Peptides corresponding to HRT1 or HRT2 were identified among 137 proteins (115 subfamilies) assigned (*Figure 2—source data 1,2*); however, this result does not allow to discriminate between these two cPTs since some detected fragments correspond to the sequences shared by both enzymes. No other prenyltransferase or enzyme in the IPP biosynthetic pathways was detected from the proteome analyses of the detergent-washed RPs, in contrast with previous proteome analyses of non-detergent-washed RPs from *H. brasiliensis*, in which several homologues of cPT (*Dai et al., 2013*) and a homologue of 3-hydroxy-3-methylglutaryl-CoA synthase (*Xiang et al., 2012*), one of late-limiting enzyme in the IPP biosynthetic pathway, were detected. However, we also identified a protein assigned as a homologous protein of NgBR (*Harrison et al., 2011*). The NgBR homologue from *H. brasiliensis* showed greater similarity to higher plant orthologues (*Figure 2C*) that have one predicted transmembrane domain in the N-terminal region (*Qu et al., 2015*; *Zhang et al., 2008*; *Brasher et al., 2015*) (*Figure 2—figure supplement 2B*). In parallel with the RP proteomics, we isolated the *Hevea* homologue of NgBR by screening REF-interacting

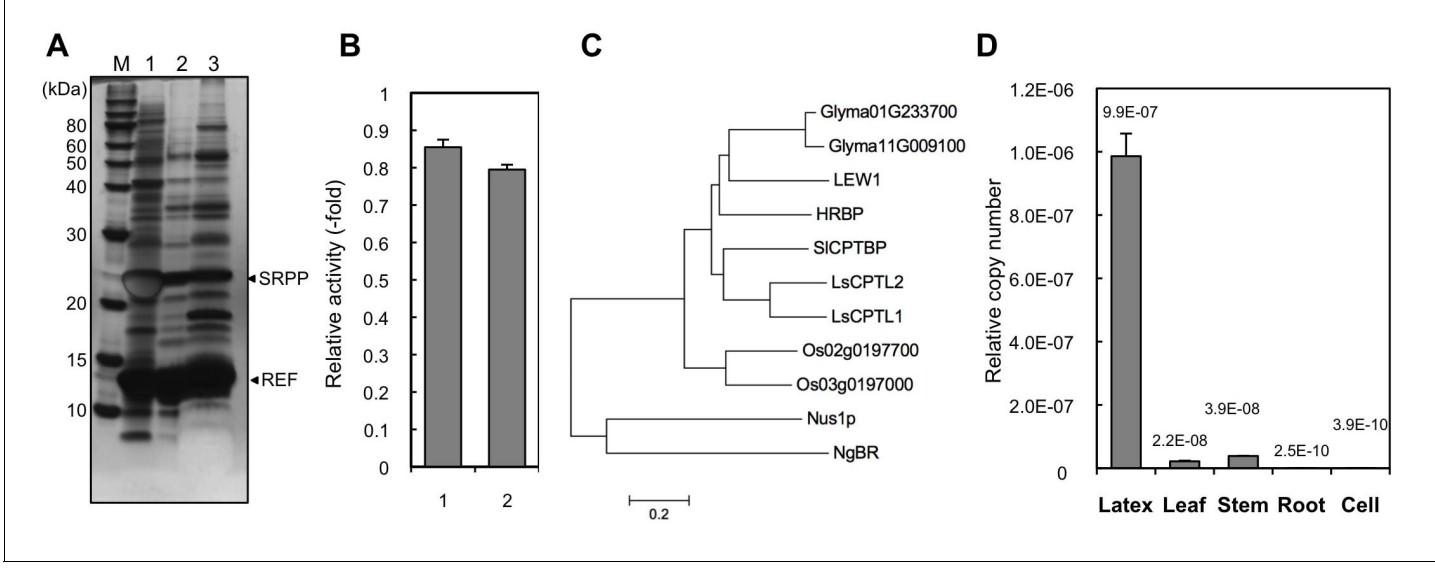

**Figure 2.** Identification of a *Hevea* NgBR homologue. (**A**) Solubilization of proteins from RPs. Proteins on RPs were solubilized by stepwise treatment with buffers containing 4 mM CHAPS (lane 1), 16 mM CHAPS (lane 2), and 7 M urea, 4 M thiourea, and 6.5 mM CHAPS (lane 3), analyzed using SDS-PAGE, and then visualized by silver staining. (**B**) Relative RTase activities of the RPs, with RTase activity of non-detergent-washed RPs set at 1.0. 1: Sustained activity on RPs after the second wash with 16 mM CHAPS; 2: activity of the second-washed RPs co-incubated with the proteins released by the two-step washes, corresponding to the mixture of samples shown in *Figure 2A*, lanes 1 and 2, after dialysis removal of the detergent. RTase activities were assayed using the standard conditions (see Materials and methods). Results are presented as the mean of three independent determinations ± SD. (**C**) Molecular phylogenetic tree of the amino acid sequences of identified NgBR family proteins. The tree was constructed using the neighbor-joining method and drawn to scale, with branch lengths in the same units as those of the evolutionary distances used to infer the phylogenetic tree. Tree members are HRBP, the NgBR homologue from *H. brasiliensis* (LC057267); LsCPTL1 from *L. sativa* (AIQ81186); LsCPTL2 from *L. sativa* (AIQ81187); SlCPTBP from *Solanum lycopersicum* (XP_004241992); LEW1 from *A. thaliana* (NP_001077518); Nus1p from *S. cerevisiae* (NP_010088); NgBR (AAI50655), and non-functionally identified NgBR-like proteins in *Oryza sativa* (Os02g0197700: NP_001046201, Os03g0197000: NP_001049268) and *Glycine max* (Glyma01G233700: NP_001242452, Glyma11G009100: XP_003537718). (**D**) Tissue specificity of the expression levels of the NgBR homologue from *H. brasiliensis*. Transcript levels were determined as copy number of *HRBP*, normalized with those of the 18S rRNA quantified from the total RNA extracted from latex, leaves, stems, roots and suspension-cultured cells of *H. brasiliensis*. Results are presented as the means of three independent determinations ± SD. RP, rubber particle.

The following source data and figure supplements are available for figure 2:

**Source data 1.** Proteins related to the natural rubber biosynthesis (upper list) and vesicular trafficking (lower list) identified in the RP proteomics.
**Source data 2.** Proteins identified in the proteomics of the detergent-washed RPs.
**Figure supplement 1.** Treatments of RPs with various detergents.
**Figure supplement 2.** Multiple alignments of amino acid sequences.

proteins against normalized latex cDNA libraries using a split-ubiquitin-based yeast two-hybrid (Y2H) system (see below). Several studies have suggested that REF may be involved in NR biosynthesis (*Berthelot et al., 2014a*), although its precise function is unclear. Quantitative analyses of the transcript levels in various tissues showed predominant expression of the NgBR homologue in latex (*Figure 2D*), along with cPTs (*HRT1* and *HRT2*) and *REF* (*Aoki et al., 2014a*), and the transcript level of the NgBR homologue in latex was comparable to those of *HRT1* and *HRT2*, but 540-fold lower than that of *REF* (*Aoki et al., 2014a*). These results suggested that the NgBR homologue might possibly form a complex with HRT1/HRT2 and REF on RPs.

## Physical interaction partnerships among HRT1, HRT2, HRBP, REF and SRPP

To further investigate specific interactions between the proteins identified by proteomic analyses, physical interaction assays using the split-ubiquitin-based Y2H system were conducted. The assays showed that the NgBR homologue could interact with HRT1 and REF, but not with HRT2 (*Figure 3*). We therefore named the NgBR homologue HRT1-REF bridging protein (HRBP). The Y2H analyses also indicated distinct REF–REF, HRBP–HRBP and REF–SRPP interactions (*Figure 3*). Neither homo-dimeric nor heterodimeric interactions for HRT1 and HRT2 were detected (*Figure 3—figure supplement 1*), in contrast with bacterial cPTs which function as homodimers (*Fujihashi et al., 2001*). A bimolecular fluorescence complementation (BiFC) assay with split mVenus protein fragments in *Nicotiana benthamiana* leaf epidermal cells revealed that these interaction partnerships were also detected *in planta*, almost independent of orientations of the mVenus fragments fused in-frame with target proteins (*Figure 4A*, *Figure 4—figure supplement 1,2*). The subcellular localizations for each BiFC signal differed among the interaction pairs. BiFC signals indicating an HRT1–HRBP interaction were observed at the Golgi, harboring or surrounding fluorescence signals for a mTurquoise2 (mTq2)-fused marker protein (XT-Golgi-mTq2), which accumulates exclusively in a medial subset of the cisternae of Golgi stacks (*Pagny et al., 2003*). HRBP–REF interactions were observed on aggregates of spherical bodies and smaller independent particles, and did not overlap with the Golgi signals, and REF–REF interactions were localized mainly on oval bodies and smaller spherical particles. HRBP–HRBP interactions were only observed at the perimeters of spherical particles. In contrast to the result obtained by the Y2H assay, distinct fluorescent signals indicating an HRT2–HRBP interaction were observed at the plasma membrane in the BiFC assay.

## Subcellular localizations of HRT1, HRT2, HRBP and REF

To get information about the detailed subcellular localizations of HRT1, HRT2, HRBP and REF, these proteins fused in-frame with mTq2 or mVenus were expressed in *N. benthamiana* leaf epidermal cells. The fluorescence signals for mTq2-fused HRBP or REF showed network-like patterns, which almost overlapped with an mCherry (mChe)-HDEL ER marker (*Nelson et al., 2007*) (*Figure 5A*), indicating that these proteins were predominantly localized at the ER. HRBP:mTq2-derived signals were also observed as small dots, which did not overlap with signals for an mChe-fused Golgi marker, N-terminal 49 amino acids of GmManI (*Saint-Jore-Dupas et al., 2006*) (*Figure 5—figure supplement 1*). We could not detect distinct fluorescence signals for HRT1:mTq2 or HRT2:mTq2 when they were co-expressed with mVenus as an introduction marker. However, when HRT1:mTq2 was co-expressed with HRBP:mVenus, high HRT1:mTq2 fluorescence signals were detected at the Golgi, with weaker signals at the ER. Additionally, co-expression of HRT1:mTq2 changed the subcellular localization of fluorescent signals for HRBP:mVenus from the ER to the Golgi, which perfectly over-lapped with those of the Golgi-localized HRT1:mTq2. These results indicate that HRT1 is stabilized in *N. benthamiana* cells by complex formation with the HRBP, which induces translocation of HRBP from the ER to the Golgi. In the co-expression system of HRT2:mTq2 and HRBP:mVenus, HRT2:mTq2-derived signals localized at the plasma membrane, showing a partial overlap with a marker protein mChe-AtPIP2A localized at the ER to the plasma membrane (*Nelson et al., 2007*). These results indicate that HRT2 is also stabilized by HRBP in *N. benthamiana* cells. Surprisingly, in this co-expression system, subcellular localization of HRBP:mVenus was clearly altered from the ER to the plasma membrane (*Figure 5A*), concordant with the BiFC signals for the HRT2–HRBP interaction at the plasma membrane (*Figure 4A*). Taken together with the basic subcellular localization of HRBP on the ER, HRBP is assumed to assist incorporation or stabilization of HRT2 on the ER membrane and to be co-translocated to the plasma membrane with HRT2. Additionally, the different subcellular localization of HRT1 and HRT2 co-expressed with HRBP indicates that they have different roles. Considering that RPs are speculated to originate from the ER or Golgi apparatus (*Chrispeels and Herman, 2000*), HRT1 may be more important in NR biosynthesis than HRT2. By contrast, the subcellular localizations of REF and HRBP were not influenced by co-expression (*Figure 5—figure supplement 2*). Although the fluorescence patterns of REF:mTq2 and HRBP:mVenus matched almost completely, formation of the REF–HRBP complex is considered to be limited to specific particles, possibly derived from the ER (*Figure 4A*). The formation of homomeric complexes of REF and HRBP only on small bodies and/or particles (*Figure 4A*) suggests that physical interactions among HRT1,

**Figure 3.** Physical interaction analyses of the proteins by a split-ubiquitin-based Y2H assay. Growth of yeast strains, harbouring genes fused with N-terminal (N*ub*) or C-terminal (C*ub*) halves of split ubiquitin fragments (listed on left side), on SD(-WL) and SD(-WLHAde). In each panel, five-fold dilution series of cultures, with cell densities adjusted to be equivalent for each strain, were spotted. P.C.: the positive control plasmid harboring Ost1-N*ub*, which is designed to activate the reporter system independent of a protein-protein interaction. –: empty vector as negative control.

*Figure 3 continued on next page*

*Figure 3 continued*

The following figure supplement is available for figure 3:

**Figure supplement 1.** Split-ubiquitin-based Y2H assays.

HRBP and REF might correlate with the protein topology on the membrane of specified compartments derived from the ER or the Golgi.

## Formation of the HRT1-HRBP-REF ternary complex on the ER

Here, we focused on the formation of the HRT1-HRBP-REF ternary complex. When REF:mCherry was co-expressed, BiFC signals for the HRT1-HRBP interaction were not observed at the Golgi, but overlapped with the network-like REF:mCherry signals (*Figure 4B*), indicating REF-oriented translocation of the HRT1-HRBP complex to the ER. As expected from the Y2H assay results (*Figure 3—figure supplement 1*), we could not detect a distinct BiFC signal indicating an HRT1-REF interaction, but could observe weak BiFC signals on aggregates of spherical bodies and smaller independent particles when HRBP without fluorescent tag was co-introduced (*Figure 5—figure supplement 3*), suggesting an HRBP-mediated interaction between HRT1 and REF. Co-expression of three proteins, HRT1:mTq2, HRBP:mVenus and REF:mCherry, revealed that the fluorescent signals for each protein were almost overlapped, showing network-like patterns (*Figure 5B*), contrasting with the fluorescent signals on the Golgi and small spherical bodies in the cells co-expressing HRT1:mTq2 and HRBP: mVenus (*Figure 5A*). On the other hands, the plasma membrane localizations of HRT2:mTq2 and HRBP:mVenus were not influenced by the co-expression of REF:mCherry localized at the ER (*Figure 5B*). Taken together, these results indicated formation of a protein complex consisting of HRT1, HRBP and REF primarily localized on the ER and ER-emerged-specific particles.

To confirm the ternary complex formation in *Hevea* latex, we conducted co-immunoprecipitations from solubilized RP-bound proteins using anti-HRBP or anti-HRT1/HRT2 antibodies. In this study, we were not able to distinguish between HRT1 and HRT2 by a Western blotting since the amino acid sequence chosen as a suitable epitope peptide to generate polyclonal antibodies was shared by two proteins (see Materials and methods). As expected from the interaction partnerships revealed by Y2H and BiFC analyses, HRT1/HRT2 and REF were co-immunoprecipitated with HRBP (*Figure 6A*). Moreover, when anti-HRT1/HRT2 antibody was utilized for immunoprecipitation, not only HRBP, but also REF was co-precipitated with HRT1/HRT2. The results suggested the formations of the ternary protein complexes (HRT1/HRT2-HRBP-REF) on RPs in vivo.

## Reconstitution of RTase on WRPs

To investigate the functional significance of the complex of HRT1–HRBP–REF on RPs in NR biosynthesis, we expressed and reconstituted these proteins on liposomes by cell-free translation-coupled introduction (*Nozawa et al., 2010*) and analyzed the RTase activity by using $^{14}$C-IPP and FPP as substrates. Most of the expressed HRT1 was incorporated into liposomes regardless of the co-existence of HRBP and REF (*Figure 7—figure supplement 1A*). However, no RTase activity was detected from any proteoliposomes (data not shown). Therefore, we developed a method for cell-free translation-coupled protein introduction onto RPs to investigate the importance of the target membrane architecture and/or supplemental protein factors. To prepare RPs with minimal endogenous proteins, 50kRPs were washed with 8 mM CHAPS for 1 hr (see Materials and methods), which was optimized to suppress irreversible coagulation of the resulting detergent-washed RPs (WRPs) during the cell-free translation reaction, probably caused by a loss of membrane-bound proteins important for RP stability. Washing in this condition resulted in removal of a lot of RP proteins, including REF and SRPP (*Figure 7—figure supplement 2A*), whereas 89% of the RTase activity was retained on the WRPs (*Figure 7—figure supplement 2B*). After the cell-free translation reaction with WRPs, the WRP with exogenously expressed proteins can be easily separated by ultracentrifugation into the top layer (see Materials and methods and *Figure 7—figure supplement 3*). Localizations of the exogenously expressed proteins on WRP were analyzed by SDS-PAGE (*Figure 7—figure supplement 1B*) and Western blot (*Figure 7*). Similar to the case of proteoliposomes, most expressed

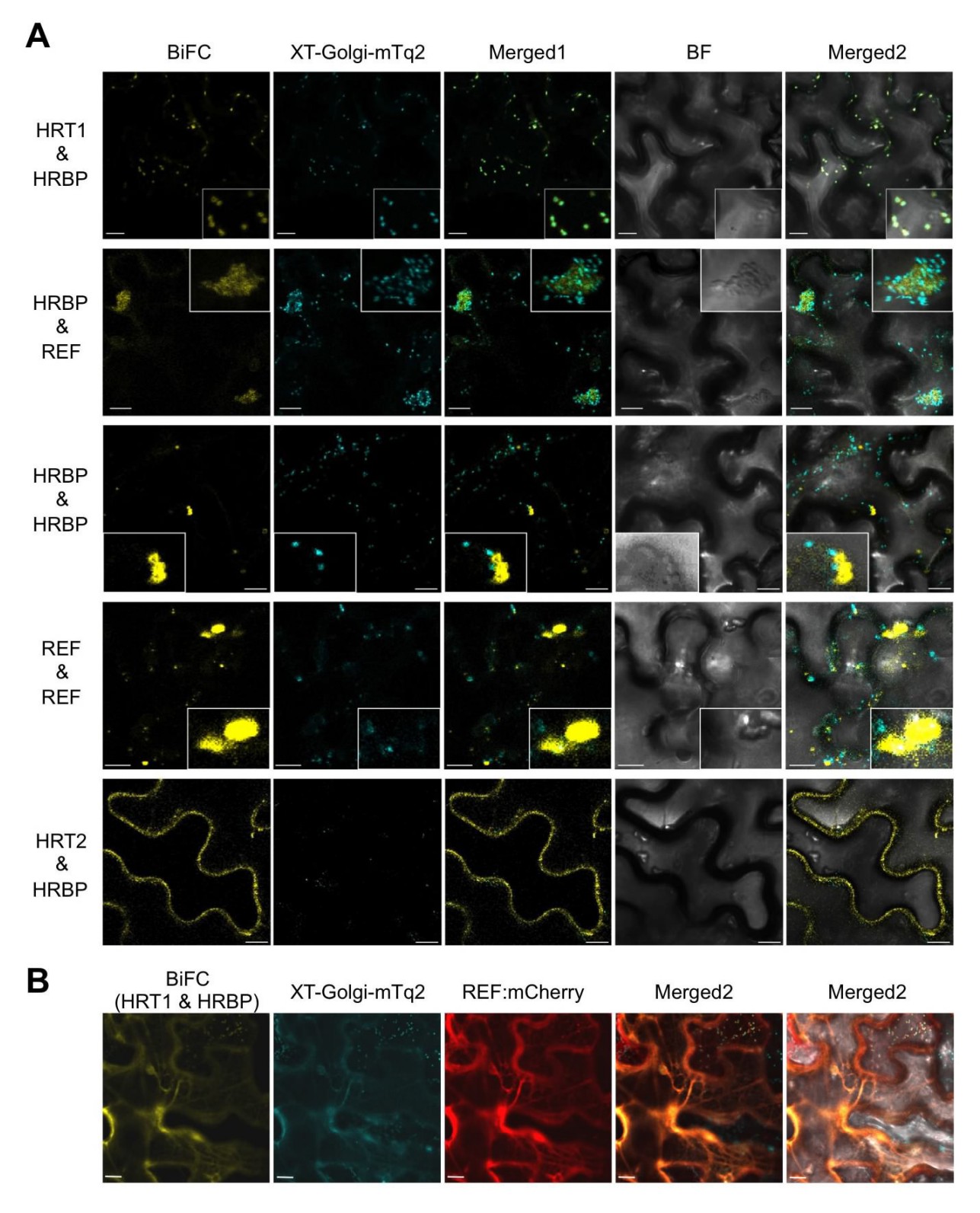

**Figure 4.** *In planta* interaction analyses of the proteins by monitoring BiFC. (**A**) Fluorescent images of *N. benthamiana* leaf epidermal cells infected with *Agrobacterium* harbouring a binary vector constructed with pDOE-07 (HRT1–HRBP, HRT2–HRBP HRBP–REF and HRBP–HRBP) or pDOE-05 (REF–REF) were observed using a confocal laser microscope. Fluorescence signals of mVenus (BiFC) and XT-Golgi-mTq2 were obtained independently and superimposed to create merged images (Merged1), which were further superimposed (Merged2) on bright-field images (BF). (**B**) BiFC assay for the
*Figure 4 continued on next page*

*Figure 4 continued*

HRT1–HRBP interaction in *N. benthamiana* epidermal cells expressing a mCherry, N-terminally fused with REF. Bars, 10 μm. REF, rubber elongation factor.

The following figure supplements are available for figure 4:

**Figure supplement 1.** BiFC assays for *in planta* interactions with split mVenus protein fragments.

**Figure supplement 2.** BiFC assays for *in planta* interactions with split mVenus protein fragments using the pDOE-05 binary vector.

proteins tested were incorporated into WRPs (*Figure 7*). When these target proteins were expressed alone, some HRT1 proteins existed in the soluble fraction, whereas HRBP and REF were incorporated almost completely into WRP. However, the amount of HRT1 not incorporated on WRPs was decreased by co-expression of HRBP and REF, indicating that these proteins promotes the incorporation of HRT1 into WRPs, but are not necessary for anchoring HRT1 on WRPs. In this system, the WRP stability was influenced by introduction of exogenously expressed proteins. Detergent washing induced RP coagulation and increased the apparent average diameter; the average diameter of RPs before the detergent treatment was 222 nm whereas that of WRPs was 1244 nm (*Figure 8*). This effect was escalated by the in vitro expression of HRBP, HRT1 and a His-tag-containing peptide from the control vector. Interestingly, however, REF expression resulted in decreased WRP coagulation, lowering the apparent average diameter towards that of the non-detergent-washed RPs, indicating that REF stabilizes WRP by preventing aggregation. This effect was dominant for the coagulation-inducible proteins, HRBP and HRT1; coagulation of WRPs was inhibited when three proteins, HRT1, HRBP and REF, were co-expressed.

The RTase assay for the exogenous protein-introduced WRPs by using $^{14}$C-IPP and FPP as substrates revealed that the introduction of HRT1 onto WRPs distinctly increased the RTase activity (*Figure 9A*), whereas HRBP or REF alone did not activate the background activity of WRPs (data not shown). The HRT1-derived RTase activity was enhanced by co-expression of HRBP and was markedly activated by additional introduction of REF. $^{14}$C-IPP incorporation in the assays of WRPs carrying three proteins was increasing linearly during 20 hr of the reaction time, whereas for WRPs carrying HRT1 only or two proteins, HRT1 and HRBP, the incorporation was weakened after 8 hr, suggesting that co-expression of REF and HRBP stabilized HRT1 on WRPs. The RTase activity of WRPs carrying three proteins depended on the HRT1 and HRBP protein levels, and the activity reached a plateau when the HRT1 and HRBP levels were apparently equivalent (*Figure 9B*). The RTase activity of the triple components showed a strict allylic substrate dependency; the RTase was most active with FPP ($C_{15}$) while the activity was low in the absence of an allylic substrate (*Figure 9C*). Other allylic substrates showed comparable activities when the triple components were used for the reactions (e.g., GPP and GGPP showed 83% and 93% activities, respectively), and the patterns of allylic substrate dependency were quite similar to those of the control reactions using WRPs expressing REF only (*Figure 9C*) and WRPs alone as reported elsewhere (*Rojruthai et al., 2010*). Gel-permeation chromatography (GPC) of the $^{14}$C-IPP-derived reaction products revealed that the size distributions of polyisoprenoids synthesized in vitro with WRPs carrying HRT1 were similar to those with control WRPs, showing unimodal size distributions with a peak at $>10^6$ as endogenous NR in WRPs (*Figure 9D*), concordant with previous reports for small rubber particles (*Rojruthai et al., 2010*) that corresponds to the 50kRP in this study. The size distributions of the HRT1-derived polyisoprenoids were not influenced by co-expression of HRBP and REF, suggesting that complex formation on the WRPs does not affect the product chain-length regulation mechanism of HRT1. These results, together with the allylic primer requirement for the RTase reaction (*Figure 9C*), suggest a possibility that HRT1 on WRP catalyzes a de novo rubber formation using allylic primer substrates, GPP, FPP and GGPP, rather than addition of IPP onto the α-terminus of pre-existing rubber molecules in WRPs, which is concordant with the backbone structure of NR, a *cis*-1,4-polyisoprene with several *trans*-isoprene units at the ω-terminus (*Tanaka et al., 1996*; *Eng et al., 1994*).

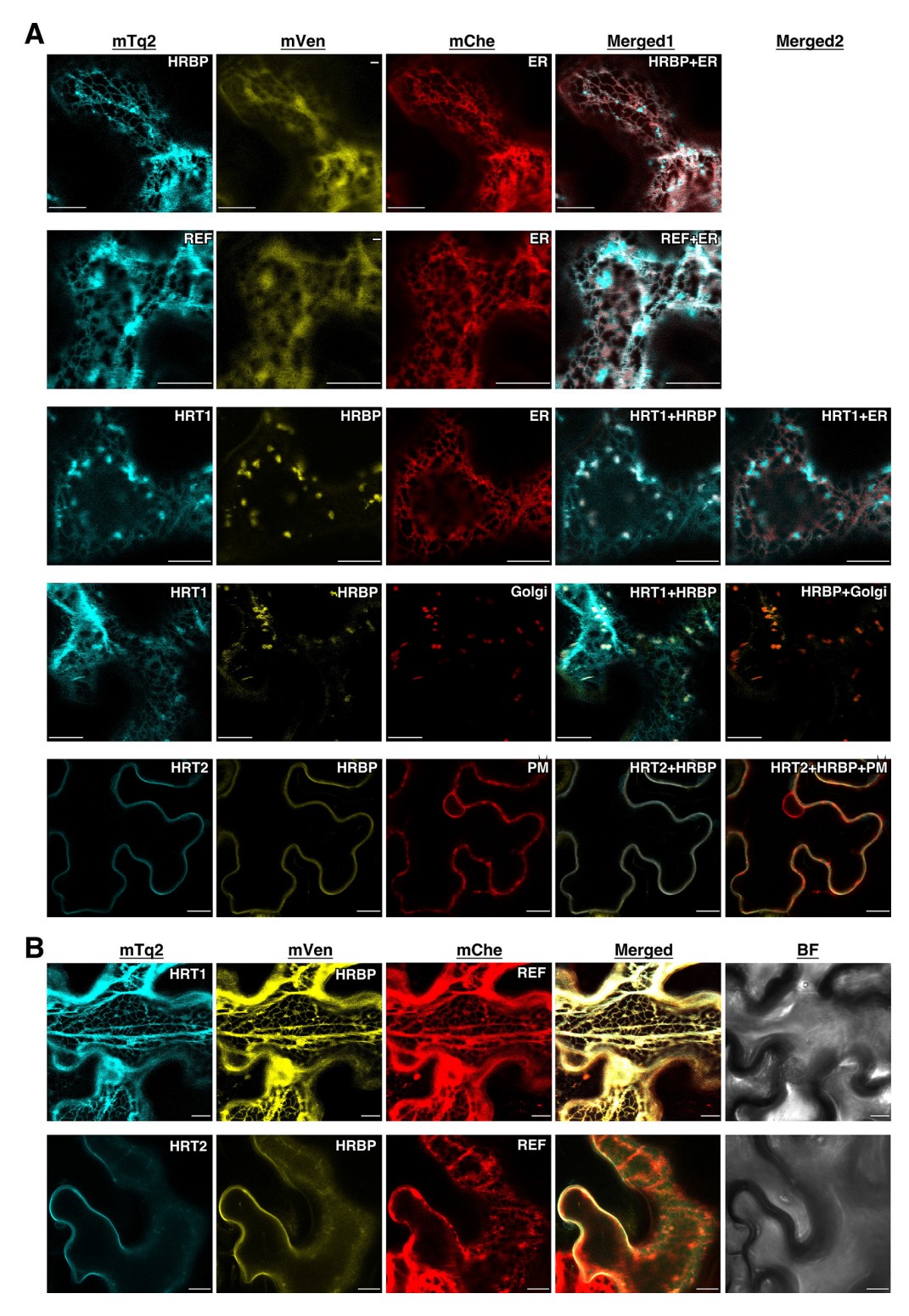

**Figure 5.** Subcellular localizations of HRT1, HRT2, HRBP and REF. (**A**) Fluorescent images of *N. benthamiana* leaf epidermal cells expressing target proteins (indicated in each panel) fused with a fluorescent protein, mTq2, mVenus or mChe, at their C-terminus. Fluorescence signals corresponding to the fluorescent proteins were obtained using a confocal laser microscope and superimposed to create merged images (Merged1 or Merged2). ER: the ER-marker (HDEL); Golgi: the Golgi marker (N-terminal 49 amino acids of GmManI); PM: the plasma membrane marker (AtPIP2A). (**B**) Fluorescent images of *N. benthamiana* leaf epidermal cells co-expressing three
*Figure 5 continued on next page*

*Figure 5 continued*

proteins, HRT1:mTq2, HRBP:mVenus and REF:mChe or HRT2:mTq2, HRBP:mVenus and REF:mChe. BF: bright-field images. Bars, 10 µm. HRBP, HRT1-REF bridging protein; REF, rubber elongation factor.

The following figure supplements are available for figure 5:

**Figure supplement 1.** Subcellular localizations of HRBP.
**Figure supplement 2.** Subcellular localizations of HRBP and REF.
**Figure supplement 3.** BiFC assays for interaction between HRT1 and REF in cells expressing HRBP.

## In vitro rubber synthesis on *H. brasiliensis* WRPs by heterologous cPT

To investigate whether the key factor for HRT1 to exhibit RTase activity is the presence of another RP-bound protein that interacts specifically with HRT1, we attempted to reconstitute the RTase activity on *Hevea* WRPs using cPT from another organism. We expressed a cPT (LsCPT3) and an

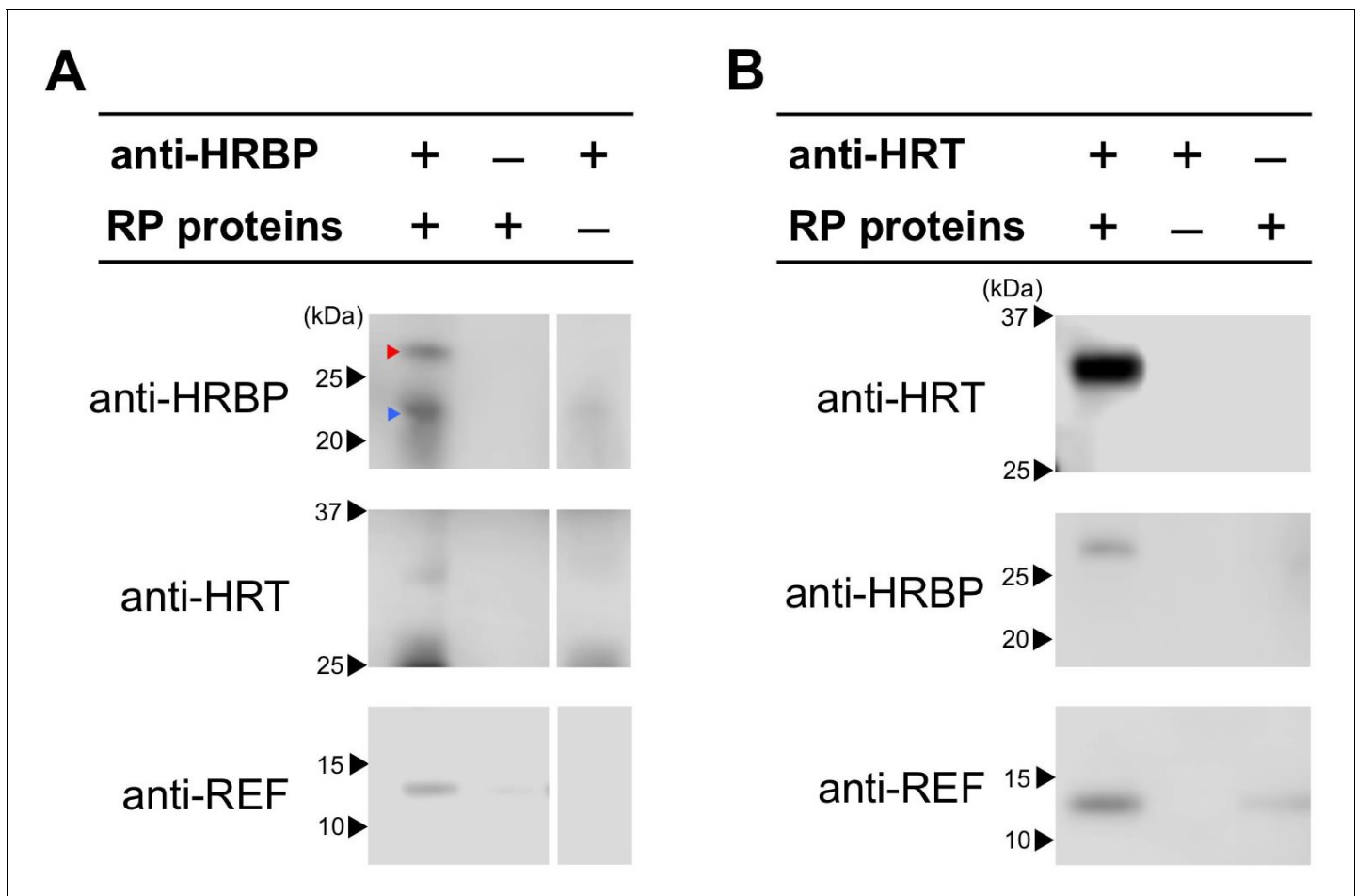

**Figure 6.** Detection of the ternary complex formation on RP by co-immunoprecipitations. CHAPS-solubilized proteins from RPs were applied for immunoprecipitation using anti-HRBP antibody (**A**) or anti-HRT1/HRT2 antibody (**B**). The immunoprecipitation experiment without an antibody or the solubilized proteins were also conducted as negative controls. Immunoprecipitated proteins were analyzed by the immunodetections with each antibody indicated. Red and blue arrowheads on the top panel of (**A**) indicate HRBP and IgG light-chains of the anti-HRBP antibody, respectively. HRBP, HRT1-REF bridging protein.

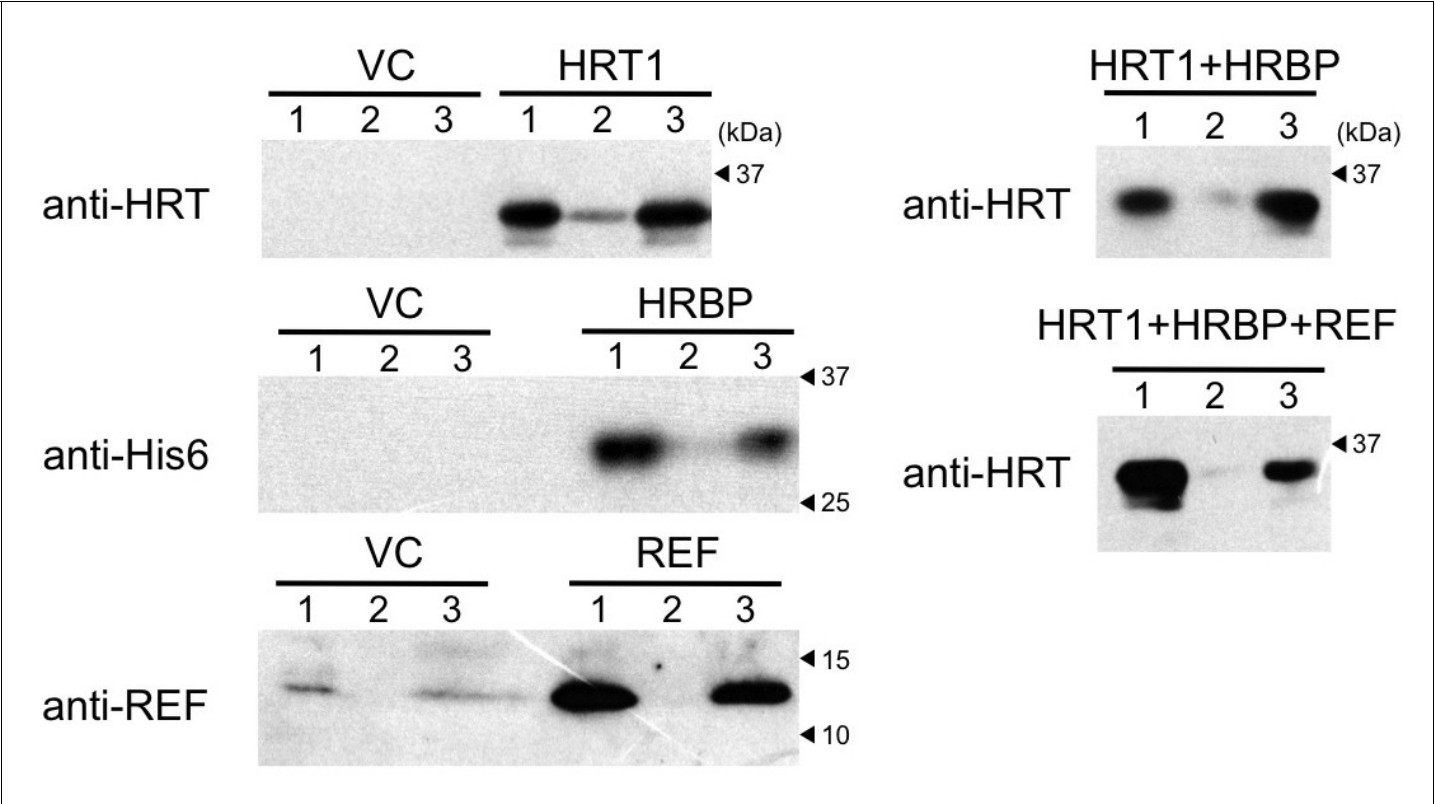

**Figure 7.** Cell-free translation-coupled protein introduction onto WRPs. Immunodetection of HRT1, His6-tagged HRBP and REF with anti-HRT1/HRT2 antibody, anti-His6 antibody and anti-REF antibody, respectively. After in vitro translation to express the protein(s) indicated at the upper part of each lane, total proteins (lane 1) and soluble (lane 2) and RP (lane 3) fractions, recovered after ultracentrifugation of the corresponding total proteins, were analyzed. VC: vector control. HRBP, HRT1-REF bridging protein; REF, rubber elongation factor; RP, rubber particle.

The following figure supplements are available for figure 7:

**Figure supplement 1.** In vitro translation of HRT1, HRT2, HRBP and REF.

**Figure supplement 2.** Sustained proteins and RTase activity on WRP applied for the cell-free translation.

**Figure supplement 3.** Workflow of protein expression on washed rubber particles with the wheat germ cell-free system.

NgBR homologue (LsCPTL2) from lettuce (*L. sativa*), which are predominantly expressed in latex but not demonstrated to be an active RTase (*Qu et al., 2015*), by the cell-free translation-coupled protein introduction system onto WRPs. LsCPT3 showed a comparable level of IPP incorporation activity in the 18-h reactions (*Figure 10A*). Interestingly, LsCPTL2 itself showed a slight enhancement of IPP incorporation activity of WRPs. Co-expression of LsCPTL2 with LsCPT3 resulted in an increase of the activity of LsCPT3 (*Figure 10A*). GPC analyses of the reaction products from the lettuce cPT on WRPs showed similar distributions of polyisoprenoids from those of HRT1 or HRT1-HRBP co-expression (*Figure 10B*), demonstrating that heterologous cPT reconstituted on the *Hevea* WRP could also exhibit distinct RTase activity. These results support our idea that the key factor for the reconstitution of RTase activity is not an unknown interacting protein for cPT but the proper introduction of cPT on the target membrane architecture of WRPs.

## Discussion

In this study, we established a novel system for cell-free, translation-coupled, hydrophobic protein introduction on detergent-washed RPs from *H. brasiliensis*, allowing de novo synthesis of NR in vitro

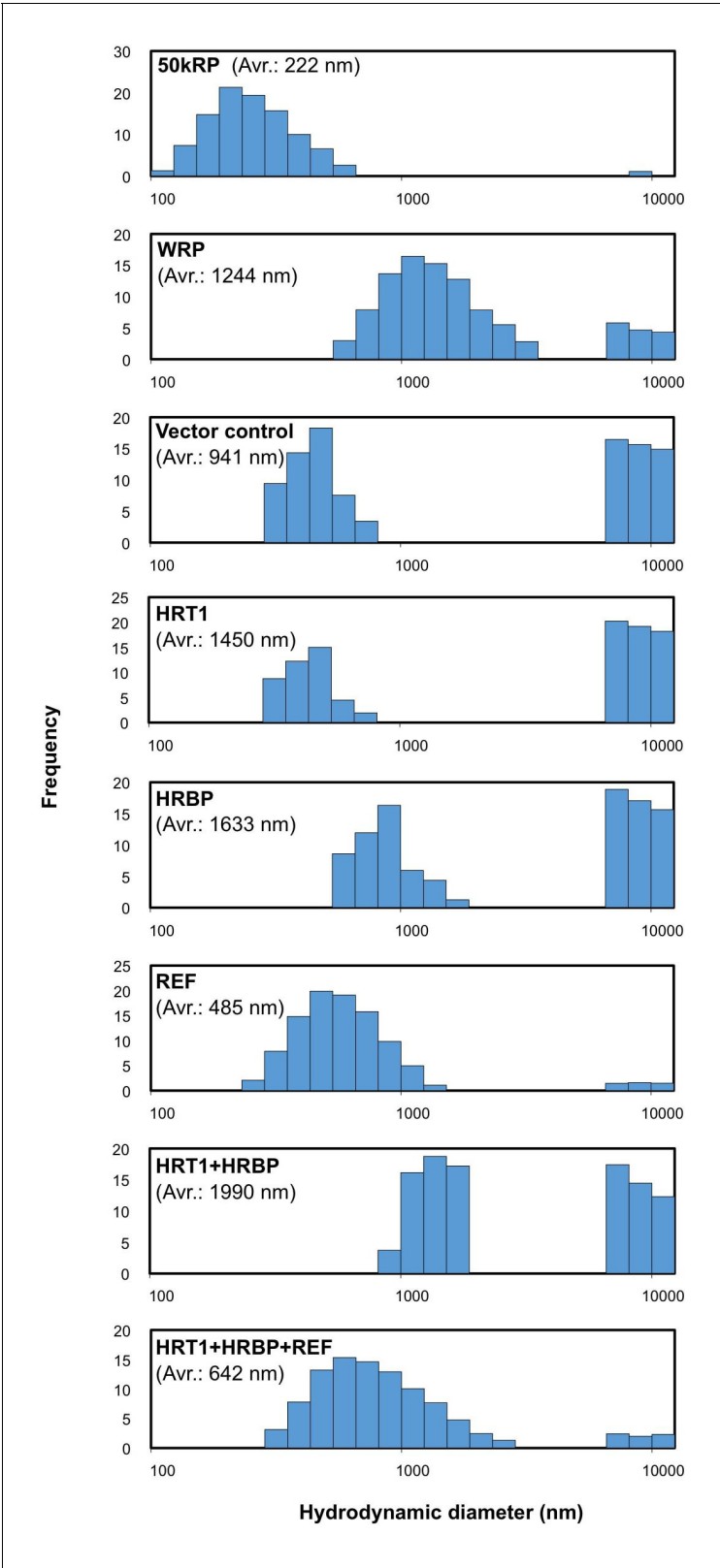

**Figure 8.** Size distributions of RPs after the in vitro translation reaction. Typical results obtained by measurement of scattering light distributions from more than three measurements by dynamic light scattering are shown. The number in parenthesis indicates the apparent average diameter of RPs. RPs, rubber particles.

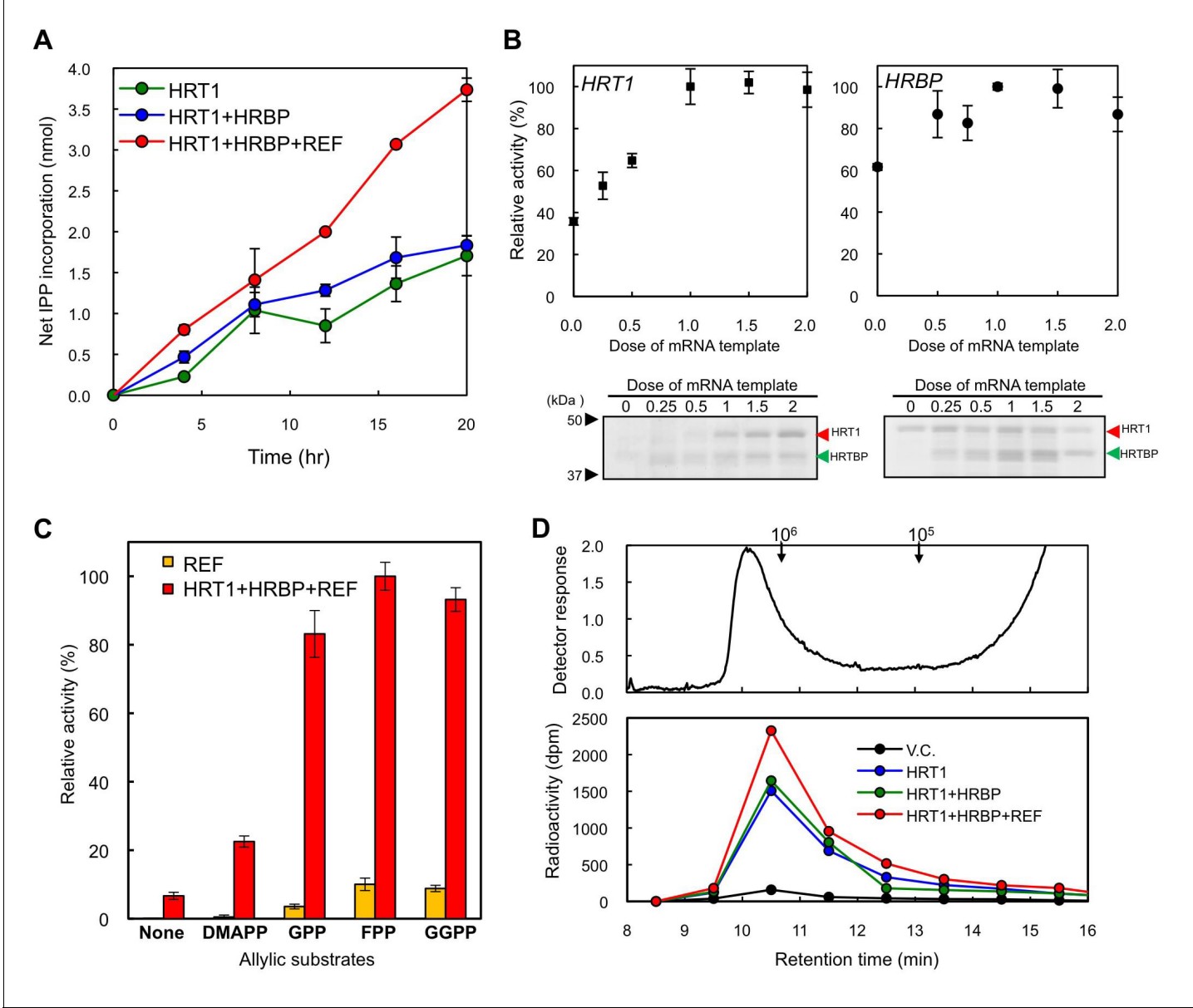

**Figure 9.** Enzymatic characterization of HRT1, HRBP and REF introduced on WRPs. (**A**) In vitro RTase activities of HRT1, HRBP and REF, introduced onto 1 μg of WRPs, in the reaction with FPP and $^{14}$C-IPP as substrates. The activity is expressed as the net IPP incorporation into the resulting rubber products, calculated by subtraction of the results with the vector control WRPs from those with the WRPs carrying the protein(s) as indicated. Results are presented as the mean of three independent determinations ± SD. (**B**) Dependencies on HRT1 and HRBP for the RTase activity of RPs carrying HRT1, HRBP and REF. Protein levels of HRT1 (left) or HRBP (right) on WRPs were varied by changing the dose of each mRNA template in the in vitro translation reaction from the optimized standard conditions (i.e. dose 1, see Materials and methods). Protein levels on WRPs were analyzed by SDS-PAGE (lower part of each assay result). Relative activities are presented as the mean of three independent determinations ± SD. (**C**) Allylic substrate dependencies of RTase activities in the 8 hr reactions with RPs carrying triple components or only REF (as a control). Allylic substrates (15 mM): dimethylallyl diphosphate (DMAPP, $C_{10}$), GPP, FPP and GGPP. (**D**) GPC analysis of reaction products from RTase assays. Molecular mass distribution of endogenous rubber molecules contained in WRPs, monitored using a refractive index detector (upper panel) and those of (*Qu et al., 2015*) C-labelled products synthesized in vitroby WRPs carrying protein(s) as indicated (lower panel). Elution peaks of commercially available polyisoprene standards (molecular weights $10^5$ and $10^6$) are indicated at the top of the upper panel. FPP, farnesyl diphosphate; GPC, gel-permeation chromatography; GPP, geranyl diphosphate; HRBP, HRT1-REF bridging protein; REF, rubber elongation factor.

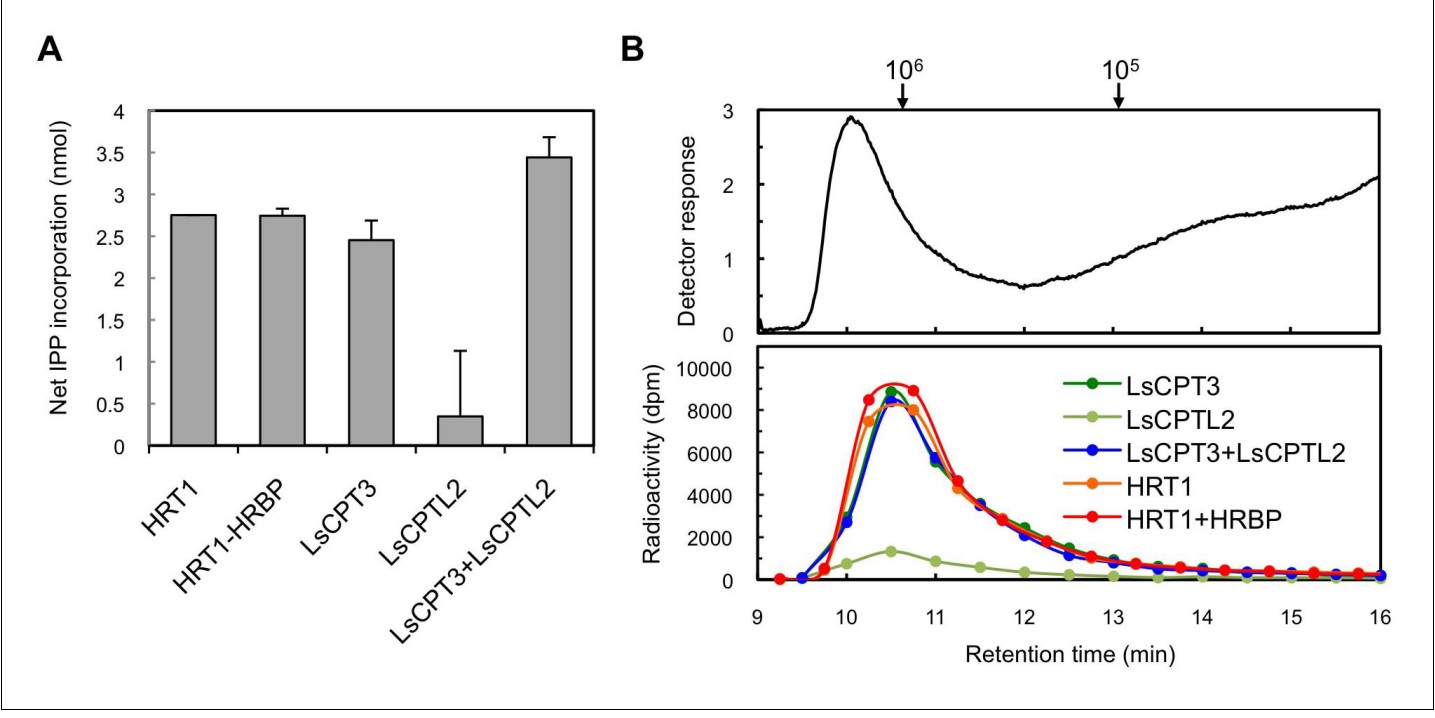

**Figure 10.** Enzymatic characterizations of LsCPT3 and LsCPT2 introduced on the Hevea WRPs. (**A**) In vitro RTase activities of LsCPT3 and LsCPT2, introduced onto 1 µg of WRPs, in the reaction with FPP and $^{14}$C-IPP as substrates. The activity is expressed as the net IPP incorporation into resulting rubber products, calculated by subtraction of results with the vector control WRPs from those with the RPs carrying protein(s) as indicated. Results are presented as the mean of three independent determinations ± SD. (**B**) GPC analyses of reaction products from the RTase assays. Molecular mass distribution of endogenous rubber molecules contained in WRPs, monitored using a refractive index detector (upper panel) and those of $^{14}$C-labelled products synthesized in vitro by RPs carrying protein(s) as indicated. Elution peaks of commercially available polyisoprene standards (molecular weights $10^5$ and $10^6$) are indicated at the top of the upper panel. FPP, farnesyl diphosphate; RPs, rubber particles.

by the recombinant protein. Additionally, we demonstrated that the key factor for reconstitution of RTase activity of latex cPTs in vitro is proper arrangement of cPT on the specific architecture of WRPs.

Since the first isolation of cPTs, HRT1 and HRT2 from *H. brasiliensis* latex, as candidates for RTase (*Asawatreratanakul et al., 2003*), several homologous cDNAs have been isolated from other NR-producing plants, *T. koksaghyz* (*Schmidt et al., 2010a*; *Schmidt et al., 2010b*), *L. sativa* (*Qu et al., 2015*) and *E. characias* (*Spanò et al., 2015*). However, practical, functional identification of RTase was not accomplished to date, owing to the absence of in vitro RTase activity distinctly exhibited by recombinant cPTs. In a previous study to elucidate activation mechanism of cPT as RTase, we showed that the refolded HRT2, which was heterologously expressed in *E. coli* and purified from its insoluble membrane fraction under denaturing conditions, was significantly activated by the addition of washed-bottom fraction (BF) of *H. brasiliensis* latex. This resulted in the formation of polyisoprenes with similar molecular size distributions as that of the NR in *Hevea* latex (*Asawatreratanakul et al., 2003*). Whereas the refolded recombinant HRT1 was not activated. BF is the sediment fraction separated by ultracentrifugation of latex and consists mainly of membrane-bound polydisperse particles called lutoids and contaminated RPs (*Moir, 1959*), with lower RTase activity compared to that of RPs (*Tangpakdee et al., 1997*; *Wititsuwannakul et al., 2003*). Therefore, it was suggested that HRT2 might function as a key enzyme in the biosynthesis of natural rubber, correlating with essential co-factor(s) included in the washed-BF. Likewise, eukaryotic long-chain cPTs (*Takahashi and Koyama, 2006*), responsible for formation of dolichols and/or long-chain ($C_{>70}$) polyprenols, such as Rer2p and Srt1p from *S. cerevisiae* (*Sato et al., 1999*; *Sato et al., 2001*), DPS from *A. thaliana* (*Cunillera et al., 2000*), and HDS from humans (*Endo et al., 2003*), do not exhibit cPT activity in vitro when expressed in *E. coli*. These eukaryotic cPTs were functionally identified by

an in vitro assay with crude membrane or microsomal proteins from an yeast mutant strain of *RER2* (*Sato et al., 1999*), expressing a heterologous cPT. This can be described as an almost solo author-ised approach for the functional identification to date. In this yeast system, both HRT2 and HRT1 show distinct cPT activity, producing polyisoprenoids with chain-lengths ranging from $C_{80}$ to $C_{100}$ as general eukaryotic long-chain cPTs. However, their catalytic properties have never been successfully modified into RTase by the addition of any latex fractions (*Takahashi et al., 2012*). These facts sug-gest that both, HRT1 and HRT2 may require additional factor(s) present within the eukaryotic cell to exhibit cPT activity and other latex-specific factor(s) to exhibit RTase activity. Additionally, HRT1/HRT2 incorporated into the yeast microsome or membrane fraction may be established as a long-chain cPT even in the presence of latex-derived factor(s). Eukaryotic post-translational modifications such as glycosylations in the ER were considered to be potential candidates for a key activation fac-tor of eukaryotic cPTs generally localised to the ER (*Shridas et al., 2003*; *Sato et al., 2001*). How-ever, this is unlikely since in this study, recombinant HRT1 and LsCPT3 showed distinct RTase activity in vitro (*Figures 9* and *10*) when they were introduced on RPs by means of the wheat germ cell-free system, in which no post-translational modifications occur without supplementation of donor sub-strates and cofactors (*Harbers, 2014*).

In our attempt to explore a candidate protein factor involved in RTase activation, we identified HRBP, an NgBR homologue from *H. brasiliensis*, as well as HRT1/HRT2, on the detergent-washed RPs (*Figure 2—source data 1*), while no other enzyme in the natural rubber or IPP biosynthetic path-ways was detected. Recently, NgBR homologs on RPs were identified from other NR producing plants, lettuce, and Russian dandelion, using an RP proteomics approach, although no cPT homolog was detected in these proteomics analyses (*Qu et al., 2015*; *Janina, 2015*). These NgBR homologs were shown to interact with cPT(s) exclusively expressed in latex. A knockdown of these NgBR homologs results in significant decrease of rubber content, suggesting involvement of the NgBR homologs in the NR biosynthesis of these NR-producing plants. These observations correlate with former reports (*Harrison et al., 2011*; *Park et al., 2014*; *Zhang et al., 2008*; *Yu et al., 2006*), in which knockdown or mutation of the NgBR family proteins from humans and *A. thaliana* resulted in severe *N*-glycosylation defects, owing to decrease in contents of dolichyl phosphates, *cis,trans*-mixed polyisoprenoids formed by the action of long-chain cPTs. While cPT is encoded by one or two genes in yeasts and animals, in higher plats it is generally encoded by a multigene family, resulting in generation of the short-, middle-, and long-chain cPTs. Short- or middle-chain cPTs from *Solanum lycopersicum*, heterologously expressed in *E. coli*, exhibit a distinct cPT activity independently in vitro, producing polyisoprenoids shorter than $C_{60}$ (*Akhtar et al., 2013*). Additionally, the recombi-nant protein of a *cis,trans*-mixed heptaprenyl diphosphate ($C_{35}$) synthase from *A. thalina*, also exhib-its a cPT activity independently without corresponding NgBR family protein (*Kera et al., 2012*; *Surmacz et al., 2014*). Taken together, the NgBR family protein is considered to be a subunit pro-tein of a long-chain cPT on the ER, responsible for the dolichol or NR formations, resulting in enhancement of cPT activity. However, it is unclear whether involvement of the NgBR family protein is a stringent requirement for cPT activation since various eukaryotic long-chain cPTs, including HRT1 and HRT2, expressed in *S. cerevisiae* show distinct activity without co-expression of a partner NgBR family protein. In vitro results in the present study emphasize that an NgBR family protein is not necessary for the catalytic activity of cPT to produce polyisoprenoids that correspond to NR; HRT1 and LsCPT3 showed a significant rubber-producing activity when introduced into WRPs (*Fig-ures 9*, *10*). Due to low amino acid sequence identities between HRT1/HRT2, and LsCPT3 and between HRBP and LsCPTL2 (53% and 45%, respectively), it is unlikely that the small amount of HRBP retained on RPs after the detergent wash could have interacted with LsCPT3 as a surrogate partner for its activation. In practice, we have analysed cross-species interaction partnerships between cPT and NgBR family proteins from different plant species by BiFC in the *N. benthamiana* system, but no distinct interaction between LsCPT3 and HRBP can be detected (unpublished data). It is interesting that coexpression of LsCPT3-LsCPTL2 on WRPs caused a slight increase in efficiency of rubber production, whereas that of HRT1-HRBP did not produce such an effect (*Figure 10A*). Whether similar phenomena are observed using other cPTs and NgBR homologs from NR-producing plants remains to be determined. Taken together with the observation that distinct fluorescent sig-nals for mTq2-fused HRT1 or HRT2 in *N. benthamiana* cells were detected only when they were co-expressed with mVanus-fused HRBP (*Figure 5*), the NgBR family protein may play an important role

in proper introduction and folding of cPT or its stabilization on the ER membrane, especially in a heterologous cell expression system, which may be conserved in the RTase in NR-producing plants.

The absence of any RTase activity by HRT1 introduced on liposomes by the cell-free translation-coupled system suggests that structural arrangements of RPs, a hydrophobic core of rubber molecules surrounded by a lipid monolayer, and/or membrane proteins on RPs may be required for HRT1 to exhibit RTase activity. It has been demonstrated that appropriate interactions between lipids and the proteins are required to facilitate accurate folding of membrane proteins during translation (*Cymer et al., 2015*). To date, various methods for cell-free protein synthesis supplemented with biological membranes or membrane-mimicking structures, such as microsomes, micelles, liposomes, bicelles and nanodiscs, have been developed as a promising strategy to obtain functional membrane proteins. However, it has been pointed out that function and activity of the resulting proteins depend critically on properties of the membrane applied, such as lipid composition, phase, tension, fluidity and curvature (*Cymer et al., 2015*). Although neither HRT1 nor HRT2 have any deduced transmembrane domains (predicted by TMHMM; http://www.cbs.dtu.dk/services/TMHMM/), these cPTs may require the membrane environment of RPs or the ER for accurate and functional folding, which is considered to correlate with incorporation of HRT1 into WRPs after cell-free synthesis (*Figure 7*). In our previous study, the refolded HRT2, heterologously expressed in *E. coli* and purified under a denaturing condition, was significantly activated by the addition of washed BF (*Asawatreratanakul et al., 2003*). In this case, the partially unfolded HRT2 might be introduced into some membrane systems in BF, microsomal membranes or contaminated RPs, and refolded to exhibit RTase activity. In the present study, we tested a reconstitution of HRT2 in the WRP-based in vitro translation system. However, the protein level was very low compared to that of HRT1 for unknown reason. Also, the WRPs with recombinantly expressed HRT2 did not show distinct RTase activity (data not shown).

Of note was that the size distributions of rubber products from HRT1 and LsCPT3 were quite similar to each other (*Figure 10B*). These results suggest that the molecular size of de novo synthesised rubber by recombinant cPTs may be determined not by catalytic properties of each cPT but by structural properties of RPs, although further investigations will be needed to elucidate the mechanism for chain-length determination of rubber products. The architecture of RP is considered to be favourable for accommodation of extraordinarily long polyprenyl products formed by sequential condensation of IPP by an RP-localized RTase (*Figure 11A*). Whereas non-NR-producing eukaryotes also have organelles with a similar structural property, i.e., an ER-derived lipid droplet consists of a hydrophobic core of lipids, triacylglycerols and steryl esters, surrounded by a lipid monolayer (*Chrispeels and Herman, 2000*). Srt1p was reported to be localized on lipid droplets in *S. cerevisiae* (*Sato et al., 2001*), and Nus1p was also detected on lipid droplets using a proteomic approach (*Grillitsch et al., 2011*), suggesting co-localization or interaction of these proteins on lipid droplets. The fact that yeast does not show RTase activity suggests that arrangement of a cPT with an NgBR family protein on an organelle with a hydrophobic core to accommodate extraordinarily long polyprenyl products may not be a determinant for modification of the cPT into active RTase.

In spite of high amino acid sequence identity between HRT1 and HRT2 (88%, *Figure 2—figure supplement 2*), these cPTs showed different subcellular localizations in the *N. benthamiana* expression system; HRT1 in the Golgi or ER and HRT2 on the plasma membrane (*Figure 5A*). While subcellular localizations of HRT1 and HRBP are altered from the Golgi to the ER by co-expression of REF, HRT2 is translocated to the plasma membrane, accompanied by HRBP regardless of REF co-expression (*Figure 5B*), suggesting failure of REF to retain the HRT2-HRBP complex in the ER. These results suggest that cell biological roles of HRT2 in laticifer may be different from those of HRT1. Cell biological roles of cPT on the plasma membrane remain to be elucidated and to the best of our knowledge, this is the first report regarding the identification of a plasma membrane-localised eukaryotic cPT. In addition, *HRT1* expression is severely restricted in the latex of *H. brasiliensis*, while *HRT2* transcripts are detected in various tissues (*Aoki et al., 2014a*). According to the functional characterizations of HRT2 heterologously expressed in *E. coli* (*Asawatreratanakul et al., 2003*) and yeast (*Takahashi et al., 2012*), HRT2 essentially possess catalytic function as cPT or possibly as RTase. Nevertheless, above-mentioned evidences suggest a possibility that HRT1 may be the principal catalytic component of RTase responsible for NR biosynthesis in *H. brasiliensis*, although we still consider a possible involvement of other cPTs including HRT2 in the NR biosynthesis. Two mechanisms for the NR biosynthesis have been proposed so far (*Rojruthai et al., 2010*): the first one is de novo

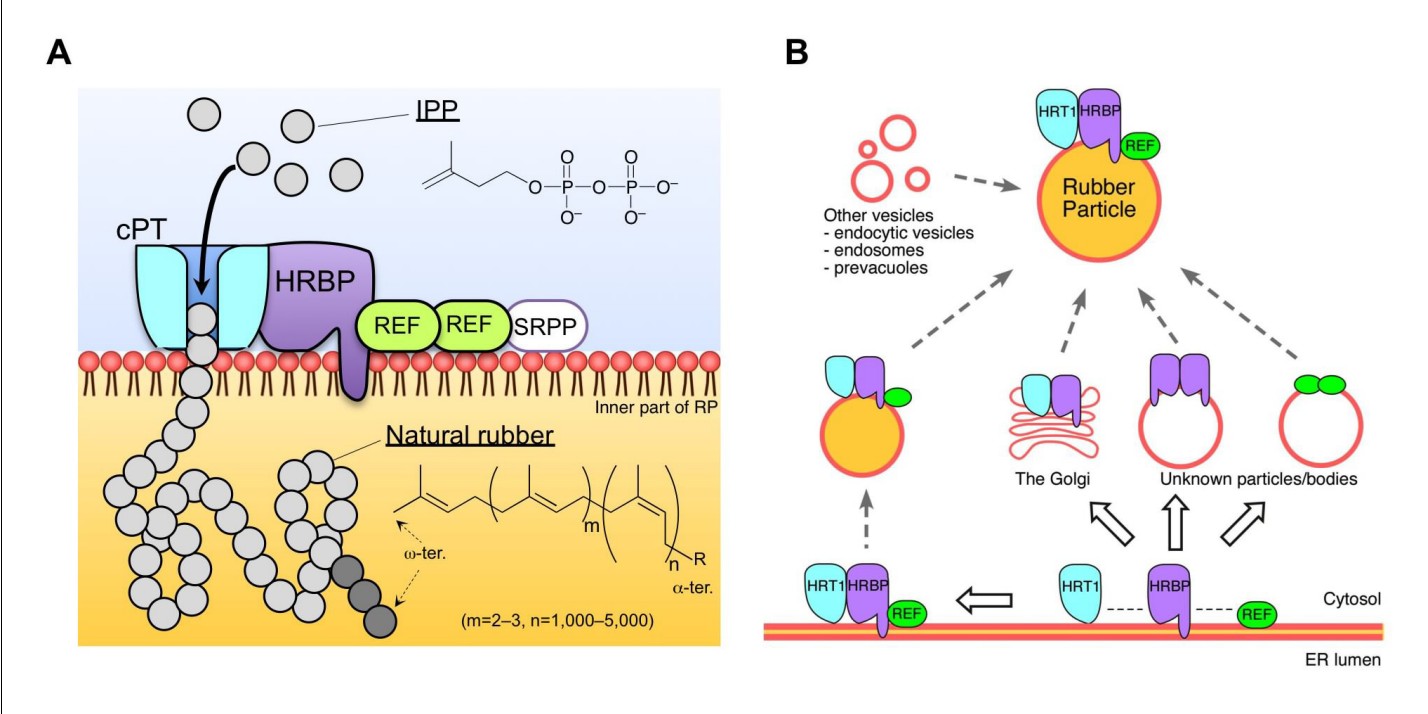

**Figure 11.** Schematic models of the rubber biosynthetic machinery on RPs (A) and RP formation correlated with the interactions of proteins (B) in the latex of *H. brasiliensis*. RPs, rubber particles.

formation of NR via sequential *cis*-1,4-condensation of IPP onto short-chain allylic diphosphates as primer substrates; the second one is elongation or addition of IPP at the $\alpha$-terminal reactive moiety of pre-exiting rubber molecules or polyprenyl diphosphates in RPs. The allylic diphosphate primer requirement for the RTase reaction (*Figure 9C*) and no detectable proteins responsible for the short-chain allylic diphosphate biosynthesis on the detergent-washed RPs (*Figure 2—source data 1*) suggest a possibility that HRT1 on WRP catalyzes a de novo rubber formation using allylic diphosphate as primer substrates. However, an alternative scenario that HRT2 on the WRPs functions as cPT to form polyprenyl diphosphates, which might be utilised by HRT1 as acceptor substrates in the latter NR biosynthetic mechanism, can not be ruled out.

For the next logical step, topologies of HRT1 and HRBP on membrane systems need to be discussed. HRBP is a distantly-related structural homolog of HRT1. The amino acid sequence of HRBP shows some similarity to that of HRT1 (sequence identity: 10.9%), although only HRBP is predicted to have a membrane-spanning region at its N-terminus. As the three-dimensional structure of HRT1 is not yet available, the most suitable templates for predicting the structure of HRBP are bacterial cPTs, which are shown to form homodimers within their crystal structure (*Fujihashi et al., 2001*). The results from the Y2H and BiFC assays show that HRBP interacts with HRT1, whereas HRT1 does not interact with itself; i.e., HRT1 may form a heteromeric complex with HRBP rather than form a homo-multimeric structure on RPs, although HRBP also exhibited self-interaction in the Y2H and BiFC assays. Reconstitution experiments with the two components on WRPs (*Figure 9B*) support this hypothesis: the reconstituted RTase activity is highest when these two components are expressed. Further experiments are needed to elucidate more precise stoichiometry of the complex.

We demonstrated that HRBP interacted with REF, a well-known RP-bound protein (*Berthelot et al., 2014a*). The results of this study also revealed the function of REF as a module of the RTase machinery. Although we have no data for the stoichiometry of an HRBP-REF complex, specific interaction partnerships, HRT1-HRBP and HRBP-REF (*Figures 3* and *4*), and probable formation of the ternary complex (*Figure 5B*, *Figure 5—figure supplement 3*) indicate that HRBP functions as a mediating protein between REF and HRT1 to form the ternary complex on RP. Additionally, only when REF was co-expressed with HRT1 and HRBP, the reconstituted RTase activity

showed a significant increase. Taken together, the ternary complex including HRT1 may function as rubber-producing machinery on RPs (*Figure 11A*). Interestingly, the amino acid sequence of SRPP, another RP-abundant protein in *H. brasiliensis*, contains a REF-homology domain (*Oh et al., 1999*). To date, several homologous proteins of SRPP have been identified not only from rubber-producing plants but also non-rubber producing plants, designated as the SRPP family or stress-related protein family (*Berthelot et al., 2014a*). However, no counterpart of REF, a shorter protein than SRPP family, has been found from other species. *Laibach et al. (2015)* reported that silencing of a *REF*-related gene *TbREF* affected the rubber production in the RP of Russian dandelion, even though this gene is rather homologous to SRPP. It is unknown whether TbREF interacts with other component(s) such as an NgBR homologue of Russian dandelion. The system for in vitro translation-coupled introduction of hydrophobic proteins on RPs, developed in this study, also enables us to shed light on molecular function of REF and SRPP on RPs. Our observations of REF in modulating diameters of WRPs (*Figure 8*) represents the first direct evidence of REF being an important protein in the WRP stability. *Berthelot et al. (2014b)* reported different behaviors of REF and SRPP on model membrane systems, where REF inserts into membranes and disrupts their integrity, whereas SRPP weakly interacts with membranes without any disruption. Since REF is likely to insert itself into RP, it is expected to interact with rubber molecules in RP. Our future plans include study of whether REF controls the turnover of RTase by interacting with the rubber molecule produced by HRT1 in a deeper region of the RP membrane.

Similarities between the structures of RP and the ER-derived lipid droplet (*i.e.* a hydrophobic core surrounded by a lipid monolayer) suggest that RPs may originate from the ER or Golgi apparatus (*Chrispeels and Herman, 2000*). This speculation was bolstered by identification of various proteins homologous to vesicular trafficking-related proteins in the ER or Golgi apparatus, on WRPs (*Figure 2—source data 1*). For example, Rab GTPases, RabB (ER to Golgi), RabD (ER to Golgi) and RabH (the Golgi), a Sar1-type coat GTPase (ER to Golgi), a YKT6 type R-SNARE (trans-Golgi network, TGN), and clathrins (TGN) (*Nielsen et al., 2008*; *Bassham et al., 2008*). However, the co-existence of marker proteins for other vesicles involved in different phases of membrane trafficking pathways, e.g. the endocytic/endosomal compartments (RabA, RabC, RabE, RabF, an ArfB-type coat GTPase and clathrins), vacuoles (RabG) and lipid-bodies (β-1,3-glucase [*Paul et al., 2014*]), on RPs suggests that RP formation and maturation may be complicated by fusion of multiple vesicles. Different subcellular localization of the protein complexes HRT1–HRBP and HRBP–REF may partly reflect the stepwise formation of the HRT1–HRBP–REF complex on RPs (*Figure 11B*). In addition, proteomics of the detergent-washed RPs did not identify any major lipid droplet proteins such as seipin, perilipins or oleosins (*Murphy, 2012*), consistent with a previous proteomics study of RPs (*Dai et al., 2013*). This implies that mechanisms for de novo formation from the ER and maturation of RPs may be different from those of lipid droplets.

In conclusion, this study represents important progress toward understanding the molecular machinery of rubber production on RP. Identification of the two partner proteins of HRT1 provides suggestions for possible formation of a ternary complex that functions as the RTase. Although the interactions between cPTs and NgBR homologs are generally assumed in other organisms, HRBP-REF interaction discovered in this study is specific to NR production. Cell biological analyses of the localization of the three components suggest that RPs can be generated by a fusion of various cellular vesicles. We demonstrated in vitro rubber production on WRP for the first time by heterologously expressing cPTs from the Para rubber tree and lettuce, although the assay system developed in this study still included a small amount of endogenous proteins sustained on WRPs. We also showed that HRBP and REF promoted efficient rubber synthesis in the same system when these two proteins were co-expressed with HRT1. Future work will include investigating the structures and functions of the two partner proteins of HRT1. In particular, the effect of REF on the alterations of specific membrane systems is a noteworthy observation. The in vitro translation-coupled protein reconstitution on WRPs is a versatile system for total elucidation of NR biosynthesis and may lead to molecular breeding of transgenic plants with enhanced NR biosynthetic activity.

## Materials and methods

### Plant materials and total RNA extraction

Fresh latex was obtained from the Para rubber trees (*H. brasiliensis* clone RRIM 600) growing on a commercial plantation in Thailand by tapping of the bark of the tree trunks, performed basically as described by *Kush et al. (1990)*. Young leaves with size from 15 to 20 cm, green stems which were 5 cm below the shoot apex and less than 1.5 cm in diameter and lateral roots less than 0.5 mm in diameter were collected, immediately frozen in liquid nitrogen and stored at –80°C until the total RNA extraction. Suspension-cultured cells derived from petioles of *H. brasiliensis* were cultured for 10 days after subculturing and harvested as described by *Aoki et al. (2014b)*. Total RNA in the cultured cells, leaves, stems and roots were extracted based on the phenol/SDS method (*Palmiter, 1974*). Total RNA in latex was extracted by using RNAiso plus (Takara Bio, Ohtsu, Japan), according to the manufacturer's instructions. Contaminating DNA was removed from the total RNA sample by treatment with DNase I (RNase-free; Takara Bio) at 37°C for 30 min.

### Preparation of RPs

RPs mainly consist of small RPs, which correspond to RPs in zone 2 layer (*Moir, 1959*), were prepared by a stepwise centrifugation. Fresh latex was centrifuged at 8000 ×g for 45 min to remove rubber cream mainly consist of coagulated rubber and large rubber particles. Resulting lower turbid aqueous fraction was collected and further centrifuged at 20,000 ×g for 45 min, and lower turbid aqueous layer was collected. Then, zone 2 RPs were collected as a top rubber layer after ultracentrifugation at 50,000 ×g for 45 min. The rubber layer was resuspended in the original volume of the fresh latex by addition of 0.1 M Tris-HCl (pH 7.5) supplemented with 2 mM dithiothreitol (DTT) (TD buffer). The resuspended RPs were ultracentrifuged again at same condition. Finally, the upper layer from the 2nd ultracentrifugation was collected and resuspended by adding the same buffer to 1.5 g (fresh weight)/ml of RP suspension. All manipulations were done on ice or at 4°C. The resultant RP suspension was designated as 50kRP and stored at −80°C until use.

### Proteome analysis of the washed RPs

To remove contaminated or attached proteins on the surface of 50kRP, CHAPS was added to a final concentration of 4 mM and the suspension was incubated for 1 hr at 4°C. After ultracentrifugation at 40,000 ×g for 45 min, the upper rubber layer was resuspended with the TD buffer with 16 mM CHAPS and incubated for 1 hr at 4°C, followed by ultracentrifugation to harvest the CHAPS-washed RPs. Proteins on the CHAPS-washed RPs were solubilized by the treatment with a solution, consisted of 7 M urea, 2 M thiourea, 6.5 mM CHAPS, 1% NP-40, 60 mM DTT and 5 mM EDTA. Proteins in each fraction were resolved by SDS-PAGE and stained by using Sil-Best Stain One silver staining kit (Nacalai Tesque, Kyoto, Japan) or Coomassie Brilliant Blue R250 (CBB).

For proteomics, the sample lane of 1D-SDS-PAGE gel stained with CBB was sliced and then further cut into three pieces to remove protein bands corresponding to well-known RP-abundant proteins, REF (14.6 kDa) and SRPP (24 kDa). In-gel trypsin digestion, LC-MS/MS analysis, followed by data analysis with MASCOT software (ver. 2.4.0, Matrix Science, Boston, MA, USA, RRID:SCR_014322) were carried out at Chemicals Evaluation and Research Institute, Japan. Proteins in the gel slices were reduced with 10 mM DTT in 100 mM NH$_4$HCO$_3$ for 30 min at 50°C and then alkylated with 55 mM iodoacetamide in 100 mM NH$_4$HCO$_3$ for 30 min at room temperature. Trypsin digestion was performed by adding 25 mM NH$_4$HCO$_3$ containing Sequencing Grade Modified Trypsin (Promega, Madison, WI), and overnight incubation at 30°C. The tryptic peptides were extracted with extraction solution (50% acetonitrile and 5% formic acid), evaporated until completely dry in a vacuum centrifuge, and then dissolved in 0.1% formic acid. The peptides were analyzed on a nanoLC-ESI-MS system (Waters, Milford, MA) coupled with Q-Tof micro (Waters). The chromatographic separation was carried out on reversed phase C18 columns [Symmetry C18, 180 μm inner diameter (ID) × 20 mm, (Waters), and 1.7 μm BEH 130 C18, 100 μm ID × 25 mm (Waters)] at a flow rate of 300 nl/min. The mobile-phase linear gradient underwent 1 min initial isocratic elution with 1% solvent B (0.1% formic acid in acetonitrile), followed by 1–50% solvent B for 120 min. It was then isocratically eluted with 85% solvent B for 24 min. The LC/MS/MS experiments were performed using 'survey scan' mode of the MassLynx software (Waters, RRID:SCR_014271). All the MS/MS spectra data were

deconvoluted using the MassLynx software (Waters) and transferred to peak list files for querying using the MASCOT software (Matrix Sciences) against publicly available databases of NCBI nr (RRID: SCR_003257) and Swiss-Prot/UniProtKB (RRID:SCR_004426), and a predicted protein database of *H. brasiliensis*, prepared from the leaf transcriptome data of *H. brasiliensis* [67268 contigs (Accession Number: JT914190–JT981478) (*Rahman et al., 2013*)].

## cDNA cloning of *HRBP*

A deduced coding sequence of *HRBP* was amplified by reverse-transcription PCR from the leaf total RNA using PrimeScript 1st cDNA Synthesis Kit (Takara Bio, Ohtsu, Japan) and KOD-Plus-Neo (Toyobo, Osaka, Japan) with a primer set, HbNgBR-Fw and HbNgBR-Rv (source data). Amplified products were analyzed by agarose gel electrophoresis and purified from the agarose gel using UltraClean15 DNA Purification Kit (MO BIO Laboratories, Carlsbad, CA). After the addition of 3' A-overhangs using a Mighty TA-cloning Reagent Set for PrimeSTAR (Takara Bio), the cDNA was subcloned into pGEM-T Easy Vector (Promega) and sequenced.

## Quantitative real-time reverse transcription PCR

After quantification of total RNA isolated from each tissue by using a NanoDrop Spectrophotometer (Thermo Fisher Scientific, Yokohama, Japan), reverse transcription was performed using the Prime-Script RT reagent Kit (Perfect Real Time) (Takara Bio) as the manufacturer's instructions. Real-time quantitative PCR were then run on the Eco Real Time PCR System (Illumina, San Diego, CA) using Fast SYBR Green Master Mix (Thermo Fisher Scientific, Waltham, MA), with the following gene-specific primers: HRBP-Fw and HRBP-Rv for *HRBP* and 18SrRNA-S and 18SrRNA-A for 18S rRNA, which was analyzed as the housekeeping gene. Data were analyzed using Eco Software (ver. 3.0, Illumina). A standard curve was generated using a serial dilution of plasmid harboring each gene.

## Split-ubiquitin-based Y2H library screening and interaction assay

Y2H screening with a split-ubiquitin-based system were performed to identify REF-interacting proteins from a latex cDNA library by using DUALmembrane Kit 3 (Dualsystems Biotech, Zurich, Switzerland). Full-length coding sequence (CDS) for REF was amplified by PCR using KOD-Plus-Neo (Toyobo) with a set of primers (REF-Fw and REF-Rv) and the plasmid prepared previously (*Aoki et al., 2014a*) as template, to introduce *Sfi* I recognition sites at the 5'- and 3'-ends. The resulting cDNA was purified, and subcloned into pGEM-T Easy for sequencing as described above. For construction of a bait plasmid to express REF C-terminally fused with the C-terminal half of ubiquitin (*Cub*) and a LexA-VP16 hybrid transcription factor, the *Sfi* I-digested fragment of *REF* was gel-purified and subcloned at the corresponding *Sfi* I sites of pBT3-SUC, resulting in a plasmid pBT3-SUC-REF.

For construction of a prey plasmid library harbouring latex cDNA, the total RNA extracted from latex was purified using PolyATtract mRNA Isolation System III/IV (Promega) to obtain poly (A) mRNAs. Because a limited number of genes, such as *REF* and *SRPP*, are expressed at exceptionally high levels in laticifers (*Aoki et al., 2014a*; *Ko et al., 2003*; *Han et al., 2000*; *Chow et al., 2007*), the poly (A) mRNAs were reverse transcribed, and subsequently normalized using EasyClone normalized cDNA library construction package (Dualsystems), according to the manufacturer's instructions. The *Sfi* I-digested fragments of the normalized cDNA library were subcloned into pPR3-N, which can express a protein N-terminally fused with the N-terminal half of the ubiquitin mutant I13G (N*ub*G).

S. *S. cerevisiae* strain NMY51 (*MATα his3Δ200 trp1-901 leu2-3,112 ade2 LYS2::(lexAop)₄-HIS3 ura3::(lexAop)₈-lacZ ade2::(lexAop)₈-ADE2 GAL4*) was transformed with pBT3-SUC-REF, using Yeast-maker Transformation System 2 (Takara Bio), according to the manufacturer's guidance, and selected on the synthetic dropout (SD) medium lacking Trp and Leu [SD(-WL)] at 30°C. Resulting transformants grew well on SD(-WL) were isolated, cultured and then transformed again with the pray library. Approximately $9 \times 10^5$ transformants were screened on the SD medium lacking Leu, Trp, His and adenine [SD(-WLHAde)]. Survived strains were isolated, and applied for the second screening on SD(-WLHAde) supplemented with 1–5 mM 3-amino-1,2,4-triazole (3-AT) (Sigma-Aldrich, St. Louis, MO). Finally, survived strains in the second screening were isolated, and applied for colony PCR to amplify a insert cDNA harbored in pPR3-N, using KOD FX Neo (Toyobo), and a

primer set; pPR3N Sequence Forward Primer and pPR3N Sequence Reverse Primer. The PCR-amplified cDNAs were purified, and subcloned into pGEM-T Easy for sequencing.

To analyze interactions among HRT1, HRT2, HRBP, REF and SRPP, each cDNA was amplified by PCR, purified and introduced into the *Sfi* I-site of pBT3-SUC or pPR3-N, as described above, with sets of primers listed in source data. Yeast strains harboring both bait and pray vectors were selected on SD(-WL), and cultured in the SD(-WL) liquid medium at 30°C for an overnight. Each culture was diluted with the liquid medium to an optical turbidity at 600 nm of 1, 0.2, 0.04 and 0.008, spotted on plates of SD(-WL), SD(-WLH), SD(-TLHA) and SD(-TLHA) supplemented with 1 mM 3-AT, and grown at 30°C.

## BiFC assay and subcellular localization analyses

Binary vectors for BiFC, pDOE-05 and pDOE-07 (*Figure 4—figure supplement 1*), and that for subcellular localization analyses, pDOE-13 (*Gookin and Assmann, 2014*), and the organelle marker proteins (*Nelson et al., 2007*) were obtained from Arabidopsis Biological Resource Center (ABRC, The Ohio State University, USA, RRID:SCR_008136). For *REF* and *HRBP,* full-length CDSs were amplified by PCR with following primer sets; MCS3-REF-RsrII_Fw and MCS3-REF-AatII_Rv for *REF*, and MCS3-HRBP-RsrII_Fw and MCS3-HRBP-AatII_Rv for *HRBP,* to attach *Rsr* II *and Aat* II recognition sites at 5'- and 3'-ends, respectively, and then each *Rsr* II-*Aat* II-digested CDS was introduced into *San* DI–*Aat* II sites in the multicloning site (MCS) 3 of each vector. To introduce a gene of interest into MCS1 of each vector, full-length CDSs for *HRT1, HRT2, REF*, and *HRBP* were amplified by PCR with following primer sets; MCS1-REF-NcoI_Fw and MCS1-REF-SpeI_Rv for *REF*, MCS1-HRT1-NcoI_Fw and MCS1-HRT1-SpeI_Rv for *HRT1/HRT2*, and MCS1-HRBP-NcoI_Fw and MCS1-HRBP-SpeI_Rv for *HRBP*, to attach *Nco* I and *Spe* I recognition sites at 5'- and 3'-ends, respectively, and then each *Nco* I-*Spe* I-digested fragment was ligated into corresponding restriction enzyme recognition sites in MCS1. To construct a binary vector for expression of REF C-terminally fused with mCherry, pDOE-13-1 harboring REF at MCS1 was digested with *Sma* I and *Swa* I, and then self-ligated, and the resulting plasmid was digested with *Bam* HI and *Xba* I, and ligated with a *Bam* HI-*Xba* I-digested mCherry fragment, amplified by PCR with a primer set (mCherry-BamHI_Fw and mCherry-XbaI_Rv). To construct a binary vector for expression of HRBP and mCherry, pDOE-13 was digested with *Bam* HI and *Xba* I to remove mTq2, and then ligated with the *Bam* HI-*Xba* I-digested mCherry fragment. The resulting plasmid was digested with *Bsp* EI and *Swa* I and self-ligated after a blunting treatment to remove a mVenus fragment, followed by introduction of a *Rsr* II-*Aat* II fragment of HRBP, which was amplified by PCR with a primer set (MCS3-HRBP-RsrII_Fw and MCS3-HRBP-AatII_Rv), at it's *San* DI-*Aat* I sites.

*Agrobacterium tumefaciens* GV3101(pMP90) was grown overnight at 28°C in a liquid LB medium containing 100 μg/ml rifampicin and 25 μg/ml gentamicin. The *E. coli* strain harboring pRK2013 was grown at 37°C overnight in an LB medium containing kanamycin (50 μg/ml). Donor strain DH5α harboring each binary vector, derived from pDOE-05, pDOE-07, and pDOE-13, were also grown at 37°C in a liquid LB medium containing 50 μg/ml kanamycin. Cells were, respectively, collected by centrifugation, washed with LB medium without antibiotics, and resuspended in the original volume of LB medium. Fifty-microlitres each of cell suspensions were mixed and plated on an LB-agar plate containing 100 μg/ml rifampicin, 25 μg/ml gentamicin and 50 μg/ml kanamycin to select *Agrobacterium* harboring a binary vector.

Transgenic *Agrobacterium* strains were cultured in a liquid LB medium, containing 100 μg/ml rifampicin, 25 μg/ml gentamycin, 50 μg/ml kanamycin, 10 mM MES (pH 5.7 with KOH), 20 μM acetosyringone, for 24 hr at 28°C. The cells were collected by centrifugation, washed thrice with an infiltration buffer [10 mM MES (pH 5.7 with KOH), 10 mM MgCl$_2$, 200 μM acetosyringone], resuspended in the same buffer to OD$_{600}$ of 0.05, and incubated at room temperature for 4–6 hr before agroinfiltration. Abaxial side of leaves of *N. benthamiana*, grown in chambers with a long-day light condition (16 hr light/8 hr dark) at 28°C for 2–4 weeks, were infiltrated with the *Agrobacterium* culture, and grown for 2 days.

For collecting confocal images, a Leica TCS SP8 confocal microscope system (Leica Microsystems, Wetzlar, Germany), equipped with a white light laser, argon laser and HyD detectors, was used. For detection of fluorescence signals derived from mVenus, mTq2 and mCherry, the excitation wavelength was 514 nm, 458 nm and 594 nm, respectively, and emission signals of 520–560 nm, 460–490 nm and 600–640 nm, respectively, were recorded using HyD detectors. At the same time, transmission images were recorded using a photomultiplier tube (PMT)-type detector.

## Protein expression with the wheat germ-cell-free system

HRT1 and its partner proteins (HRBP and REF) were synthesized with the wheat germ-cell-free system. The full-length CDSs for *HRT1*, *HRS*, and *REF* were amplified by PCR with the following primer sets; CF-HRT Fw and CF-HRT Rv for *HRT1*, CF-REF Fw and CF-REF Rv Stop for *REF*, and CF-HRBP Fw and CF-HRBP Rv for *HRBP*. The PCR products were purified, and subcloned into pGEM-T Easy (Promega) for sequencing, resulting in pGEM-HRT1, pGEM-REF, and pGEM-HRBP. The cDNA fragment of *HRT1* was digested from pGEM-HRT1 with *Bam* HI–*Not* I, and cloned into the corresponding sites of pEU-E01-His-TEV-MCS-N2 (CellFree Sciences, Matsuyama, Japan), resulting in pEU-N2-HRT1. The DNA fragments of *HRBP* and *REF* were digested with *Xho* I, and *Xho* I-*Bam* HI, respectively, and cloned into the corresponding site(s) of pEU-E01-His-TEV-MCS-C1 (CellFree Sciences) resulting in pEU-C1-HRTBP and pEU-C1-REF, respectively. The full-length CDSs for *LsCPT3* (Accession No. KF752488) and *LsCPTL2* (Accession No. KF752485) were chemically synthesized (Gene-Script Japan, Tokyo, Japan) and subcloned into the *Xho* I-*Kpn* I sites of pEU-E01-His-TEV-MCS-N2 and the *Eco* RV-*Xho* I sires of pEU-E01-His-TEV-MCS-C1, respectively.

For in vitro transcription and translation, WEPRO7240H Expression kit (ENDEXT Technology, Cell-Free Sciences) was used. Messenger RNA was synthesized in vitro using reagents, Transcription buffer LM, NTP mix and SP6 RNA polymerase, provided from the kit, and the plasmid described above or their empty vector as a template. The synthesised mRNA was purified by ethanol precipitation, and the mRNA pellet was resuspended in 25 μl of SUBAMIX SGC buffer (ENDEXT, CellFree Science). The volume of the mRNA solution added in the standard in vitro translation reaction were varied among each component as follows; HRT1, 3.75 μl; HRBP and REF, 0.75 μl, which were optimized to make the resulting protein expression levels almost even and to minimise negative effects on RP stability by the introduction of these exogenous proteins.

Liposomes (20 mg/ml in a SUBAMIX SGC buffer) were prepared from L-α -phosphatidylcholine (Soy PC, Avanti Polar Lipids, Inc. Alabaster, USA) by sonication for 5 min on ice. The sonicated liposomes were then treated by Mini-extruder (Avanti Polar Lipids, Inc) equipped with a polycarbonate membrane of 0.1 μm pore size. WRPs were prepared by incubating 50kRP with 8 mM CHAPS for 1 hr at 4℃ followed by ultracentrifugation at 50,000 ×g for 30 min. The upper rubber layer was resuspended with a SUBAMIX SGC buffer.

Proteins were synthesized in a reaction mixture (50 μl) contained 30 mM HEPES-KOH (pH 7.8), 100 mM potassium acetate, 2.7 mM magnesium acetate, 0.4 mM spermidine, 2.5 mM DTT, 1.2 mM ATP, 0.25 mM GTP, 0.3 mM of each amino acid, 16 mM creatine phosphate, creatine kinase (0.4 mg/ml), wheat germ extract (final concentration, 60 $A_{260}$ units; WEPRO 7240H, CellFree Science), the suitable amount of mRNA, and 5 μl of WRPs or liposomes, incubated at 26℃ for 26 hr. After the reaction, the mixture of cell-free reaction was ultracentrifuged at 50,000 ×g for 20 min due to the partial purification of RPs. Then, the lower layer, a clear solution that contained wheat germ extract, was removed and the upper layer (RPs) was resuspended in original volume of SUBAMIX SGC buffer. Proteoliposomes were collected by centrifugation at 20,000 ×g for 20 min. The pellet was resuspended with SUBAMIX SGC buffer, and the resulting solution was sonicated for 10 s.

## Enzymatic assay of rubber synthesizing activity

The RTase activity of recombinant proteins from cell-free system was measured by the modified method of *Takahashi et al. (2012)*. The reaction mixture contains 50 mM Tris-HCl (pH 7.5), 2 mM DTT, 5 mM MgCl$_2$, 15 μM FPP, 100 μM [4-$^{14}$C]IPP (740 GBq/mol, PerkinElmer, Waltham, MA) and a suitable amount of RPs or proteoliposomes partially purified from the cell-free reaction. The reaction mixtures were incubated at 30℃ for 16 hr. After incubation, 0.2 ml of H$_2$O saturated with NaCl was added to the reaction mixture, and then polyprenyl diphosphate products, apparently smaller than natural rubber molecular weight range, were extracted by 1 ml of 1-butanol saturated with H$_2$O. The residual rubber in aqueous phase was extracted two times with 500 μl of toluene/hexane mixture (1:1 v/v) (T/H). The radioactivity of the T/H extract was measured with a liquid scintillation counter (PACKARD 1600TR) and the amount of $^{14}$C-IPP incorporated into rubber molecules was calculated as rubber synthesizing activity. For determination of the relative RP (i.e., rubber) contents in each reaction, 15 μl of an aliquot of the RP solution, partially purified after the cell-free translation reaction, was diluted into 0.3 ml of H$_2$O saturated with NaCl, and then washed with 1 ml of 1-butanol. After removing the 1-butanol layer, the residual rubber in aqueous phase was extracted two times

with 500 µl of tetrahydrofuran (THF). The absorbance at 210 nm of the THF extract was used to estimate relative rubber content in each RP solution applied for the assay, which was used for normalization of the RTase activity. The relative rubber content in the RP solution, derived from the cell-free reaction to express HRT1, was taken to be 1.0. For substrate dependencies assays, following compounds were used as an allylic substrates: dimethylallyl diphosphate (DMAPP, $C_5$), $E$-geranyl diphosphate (GPP, $C_{10}$), FPP and $E,E,E$-geranylgeranyl diphosphate (GGPP, $C_{20}$). The concentration of each allylic substrate in the assay was 15 µM.

## Analysis of reaction products from the enzymatic assay

The T/H extracts from rubber synthesizing assay were subjected to gel permeation chromatography (GPC) which was carried out with Tosoh GPC-8020 (Tosoh, Tokyo, Japan), equipped in tandem with a series of a TSK gel GPC columns, TSKguardcolumn MP(XL) and two Multipore $H_{XL}$-M columns (Tosoh). The chromatography was carried out at 40°C using THF as eluent, at a flow rate of 1 ml/min. The eluate was monitored by a refractive index detector RI-8020 (Tosoh) and subsequently collected as 1-ml fractions (1-min interval) to measure radioactivity. The molecular mass of the reaction products were estimated by comparison with the retention times of commercially available polyisoprene standards (Mw: 999000, 110000, and 9600, Polymer Standards Service, Mainz, Germany).

## Western blotting

To detect the protein expression of HRT1 and REF, western blot analyses were carried out using custom polyclonal antibodies (anti-HRT and anti-REF antibodies, Eurofins Genomics, Tokyo, Japan) with ECL-Plus Western Blotting Detection Reagents (GE Healthcare, Little Chalfont, UK). These antibodies were from a rabbit injected with conjugated peptides with specific sequences for HRT1/HRT2 and REF, respectively. The peptides used for productions of both antibodies are as follows; HRT1/HRT2: $NH_2$-C+FAKKHKLPEGGGHK-COOH, REF: $NH_2$-MAEDEDNQQGQGEG+C-COOH and $NH_2$-C+RAAPEAARSLAS-COOH. As a secondary antibody for both anti-HRT1/HRT2 and anti-REF antibodies, anti-Rabbit IgG-HRP polyclonal antibody (MBL, Nagoya, Japan) was used. For His6-tagged HRBP (His-HRBP), anti-6×Histidine monoclonal antibody (mouse, clone No. 9C11, Wako Pure Chemicals, Osaka, Japan) was used to detect its C-terminal His-tag. The secondary antibodies for the His-HRBP detection was ECL mouse IgG, HRP-linked whole ab (GE Healthcare).

## Co-immunoprecipitation

To detect the protein complexes of HRT1/HRT2, HRBP and REF on RPs, proteins on 50kRP were solubilized by 8 mM CHAPS as the WRP preparation for the wheat germ-cell-free system. The 50-µl portions of the solubilized proteins were then mixed with the custom polyclonal antibody, anti-HRT1/HRT antibody or anti-HRBP antibody, which was prepared from a rabbit injected with two conjugated peptides with specific sequences for HRBP, $NH_2$-C+YDSKGVLKTNK-COOH and $NH_2$-C+EAVEKDVLLDQKQM-COOH (Eurofins Genomics, Tokyo, Japan), and incubated on ice for 1 hr. Then 10-µl potion of nProtein A Sepharose 4 Fast Flow (GE Healthcare) was added to the solution and incubated on ice for 1 hr with gentle agitations in every 15 min. After the incubations, the solution mixture was centrifuged at 3000 ×g for 2 min followed by removal of the supernatant. Then a precipitated Sepharose gel was washed by 1 ml of phosphate buffered saline with 0.05% Tween 20 and centrifuged at 3000 ×g for 2 min. This wash step was repeated at least three times. After the wash, the Sepharose gel was mixed with a sample buffer for SDS-PAGE and heated at 98°C for 5 min. The heated solution was then centrifuged at 10,000 ×g for 2 min, and the supernatant was analyzed by Western blotting described in above section.

## Dynamic light scattering (DLS)

RPs purified from the cell-free reaction solutions were subjected to DLS to determine the size distribution and the averaged diameter. The RP suspension was diluted into TD buffer (1:50 dilution) and measured by using ELSZ-1000 Zeta-potential &Particle size analyzer (Otsuka Electronics, Osaka, Japan).

## Acknowledgements

We thank Dr Akira Nozawa, Ehime University, for technical support with the cell-free expression system. We are grateful to Dr Haruyuki Ishii and Dr Daisuke Nagao, Tohoku University, for technical support with the DLS system. We deeply thank Dr Kazuo Shinozaki, RIKEN, for discussing and critical reading of the manuscript.

## Additional information

### Funding

No external funding was received for this work.

### Author contributions

SY, Conception and design, Acquisition of data, Analysis and interpretation of data, Drafting or revising the article; HY, ST, Conception and design, Acquisition of data, Analysis and interpretation of data, Drafting or revising the article, Contributed unpublished essential data or reagents; TW, Acquisition of data, Analysis and interpretation of data, Drafting or revising the article, Contributed unpublished essential data or reagents; YA, MM, FY, TI, AF, Acquisition of data, Analysis and interpretation of data; YT, Conception and design, Analysis and interpretation of data, Drafting or revising the article, Contributed unpublished essential data or reagents; YM-I, Acquisition of data, Analysis and interpretation of data, Contributed unpublished essential data or reagents; KF, Conception and design, Analysis and interpretation of data, Contributed unpublished essential data or reagents; TN, Conception and design, Analysis and interpretation of data, Drafting or revising the article

### Author ORCIDs

Seiji Takahashi, http://orcid.org/0000-0002-2288-4340

## Additional files

### Supplementary files

• Supplementary file 1. List of primers used in this study.

### Major datasets

The following dataset was generated:

| Author(s) | Year | Dataset title | Dataset URL | Database, license, and accessibility information |
| --- | --- | --- | --- | --- |
| Takahashi S, Nakayama T | 2015 | Hevea brasiliensis HRTBP mRNA for transferase binding protein | http://getentry.ddbj.nig.ac.jp/getentry/na/LC057267/?format=flatfile&filetype=html&trace=true&show_suppressed=false&limit=10 | Publicly available at the DNA Data Bank of Japan (accession no: LC057267) |

The following previously published datasets were used:

| Author(s) | Year | Dataset title | Dataset URL | Database, license, and accessibility information |
| --- | --- | --- | --- | --- |
| Qu Y, Chakrabarty R, Tran HT, Kwon EJ, Kwon M, Nguyen TD, Ro DK | 2015 | Lactuca sativa cis-prenyltransferase 3 mRNA, complete cds. | http://www.ncbi.nlm.nih.gov/nuccore/KF752488 | Publicly available at the NCBI Nucleotide (accession no: F752488) |

| | | | | |
|---|---|---|---|---|
| Rahman AY, Ushar-raj AO, Misra BB, Thottathil GP, Jayasekaran K, Feng Y, Hou S, Ong SY, Ng FL, Lee LS, Tan HS, Sakaff MK, Teh BS, Khoo BF, Badai SS, Aziz NA, Yuryev A, Knudsen B, Dionne-Laporte A, McHunu NP, Yu Q, Langston BJ, Freitas TA, Young AG, Chen R, Wang L, Najimudin N, Saito JA, Alam M | 2013 | TSA: Hevea brasiliensis contigxxxxx, mRNA sequence. | https://www.ncbi.nlm.nih.gov/nuccore/?term=PRJNA82895 | Publicly available at the NCBI BioProject (accession no: PRJNA828950) |
| Qu Y, Chakrabarty R, Tran HT, Kwon EJ, Kwon M, Nguyen TD, Ro DK | 2016 | Lactuca sativa cis-prenyltransferase-like 2 mRNA, complete cds | http://www.ncbi.nlm.nih.gov/nuccore/687843842/ | Publicly available at the NCBI Nucleotide (accession no: KF752485) |

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
