## [Decision Letter]

Thank you for submitting your article "Identification and reconstitution of the rubber biosynthetic machinery on rubber particles from *Heveabrasiliensis*" for consideration by *eLife*. Your article has been reviewed by three peer reviewers, and the evaluation has been overseen by Kazunori Okada as Reviewing Editor and Michael Marletta as the Senior Editor. The following individuals involved in review of your submission have agreed to reveal their identity: Ewa Swiezewska (Reviewer #3).

The reviewers have discussed the reviews with one another and the Reviewing Editor has drafted this decision to help you prepare a revised submission.

Summary:

The reviewers find that the paper describes de novo synthesis of NR on washed rubber particles with in vitro proofs of the protein complex HRT1-HRBP-REF, which is necessary for reconstituting the activity of long length *cis*-polyprenyl isoprenoids. This breakthrough accomplished by a newly designed experimental system using washed rubber particles coupling with wheat germ cell-free protein would have a large impact on industry, and will be a milestone in the production of the NR in vitro. This is basically a nice work worth for publishing in *eLife*.

The experimental data presented are solid and convincing, but some parts of descriptions are not detailed enough and the conclusions over reach the data. The novelty of the manuscript is based on the valuable in vitro system designed to confirm the involvement of three proteins (HRT1, REF and HRBP) in biosynthesis of rubber with WRP, detergent-washed *Hevea*RPs. However, the description of WRP is not detailed enough to follow the results perfectly. This could be explained more clearly and precisely in the manuscript; what was the actual compositions of WRP other than RP, discriminating possible contamination of the components HRT1, HRBP and REF in the assay, including comment on other possible *cis*-preyltransferase such as HRT2. It would be very helpful for the discussion if the authors could indicate the levels of the proteins (and preferably Rase activity) retained on WRPs used for the enzyme assay.

Another substantial concern raised by one of the reviewers is that formation of the tertiary protein complex HRT1-HRBP-REF has only been demonstrated based on the results of the BiFC and enzymatic assays. If technically possible in 2 months allowed for the revision, in vivo results obtained by the pull-down assay or immunocytochemical analysis could be a convincing argument of nature of the tertiary protein complex. Alternatively, substantive comments on the probability of the tertiary protein complex would help to strengthen the manuscript. To reiterate, the data descriptions that clearly elucidate the nature of WRPs are insufficient.

There are many other suggestions to be revised for precise description of the conclusion, but most of them can be addressed by rewriting or rephrasing. Followings are listed points required for revision.

Essential revisions:

1) The result from the RTase assay in Figure 7 indicates that the addition of only HRT1 to WRPs results in the incorporation of 14C IPP into products. Does this mean that HRT1 by itself can exert RTase activity on WRPs, or that WRPs still contain unwashed HRBP and REF? I consider that information about WRPs is too limited in the manuscript, just describing, for example, "prepared by incubating 50kRP with 8 mM CHAPS for 1 h at 4^o^C" and "80% of the RTase activity was retained". It will be very helpful for discussion if the authors can indicate the levels of the proteins (and preferably RTase activity) retained on WRPs used for enzyme assay, which would be different from those shown in Figure 1. Moreover, I am not sure if the authors can conclude that the HRT1-HRBP-REF ternary complex is enough for RTase, based on the results from assay using WRPs, which might contain other proteins. The fact that LsCPT3 and LsCPTL2 reconstituted on *Hevea*WRPs exhibit RTase activity seems insufficient to deny the involvement of the other proteins in the case that such proteins from *Hevea* can interact also with the lettuce enzymes.

2) In the Introduction section, the authors describe "a precise role of NgBR in cPT activity is still unknown owing to a lack of in vitro studies on the function of NgBR family with purified proteins." This is true. In vitro assay with purified proteins was not performed in the previous studies, e.g., ref. 29 about SlCPT3 and SlCPTBP from tomato. But, does the same hold for the present work, which also does not use purified enzyme?

3) The explanation in paragraph six of the Discussion section does not make sense. Does HRT2, unlike HRT1, compete with REF on binding to HRBP? The interaction between HRT2 and HRBP looks weak according to the Y2H analysis.

4) In Figure 7 (left), why was ~40% activity detected in the absence of HRT1 mRNA even though HRT1 was clearly needed for RTase activity?

5) Some parts of description are difficult to follow, mainly because the purpose of the experiments are not clearly stated, results are not clearly explained or results are not described in the orderly way.

6) In many places, RP and detergent-washed *Hevea*RPs (WRP) are not clearly distinguished. For example in abstract, 'RTase activity enhancement observed for the complex assembled on RPs' in this case RPs must mean WRP. I see others in other places. It is better to be described distinctly. Figure 5—figure supplement 2 is a nice cartoon to follow.

7) Figure 1—figure supplement 1 would be better to be shown as a figure instead of supplement as an introduction for the people not familiar with this field.

8) Formation of the tertiary protein complex HRT1-HRBP-REF has been suggested based on the results of the BiFC and enzymatic assays, unfortunately the in vivo result is missing.

9) Classification of proteins applied in the manuscript (Figure 2—source data 1) is unclear. All identified protein subclasses should be shown in this table together with a complete list of proteins. Were any enzymes related to polyisoprenoid biosynthesis (e.g. HMGR, FPPS etc) identified by this method?

10) No data on the reconstitution of HRT2 in the RP-based in vitro translation system are presented in the manuscript. This fact has to be commented.

11) The statement concerning the stoichiometry of the HRT1-HRBP complex (1:1) is too speculative. Figure 7 presents only the mRNAs level but not the concentration of the respective proteins. Either western-blot analysis has to be presented or the comment has to be deleted/rephrased.

12) Authors should commend on the profile of the in vitro obtained products – should a unimodal distribution be expected? A bimodal distribution was observed earlier (Rojruthai et al., 2010).

13) What was the efficiency of the in vitro HRT assay? Only relative numbers are presented in Figure 7. It would be interesting to know any absolute data on rubber formation, e.g. the efficiency of IPP incorporation or the amount of rubber formed normalized per amount of RP or protein content.

14) Authors have to reshape their concept on the role of additional cis-prenyltransferases, besides HRT1, in NR formation. Moreover, what is the role of HRT2 in this context? The below sentence has to be deleted or rephrased or alternatively additional experiments are required.

Authors neglect (l.258) possible role of polyisoprenoid precursors in rubber biosynthesis and describe HRT1 as a sole cis-prenyltransferase involved in this process.

'… HRT1 on RP catalyses a de novo rubber formation using allylic primer substrates, GPP, FPP and GGPP, rather than addition of IPP onto the α-terminus of pre-existing rubber molecules in RPs…'

Interestingly, no rubber formation was observed for the liposome-based system despite the fact that all three proteins HRT1-HRBP-REF were present. In the opinion of this reviewer the Author should provide experimental data to support this statement showing that:

1) no preexisting polyisoprenoids were present in their RP preparation – what is known about the lipid composition of the monolayer surrounding the RB?

2) it should be shown that RPs used in the assay do not contain any protein that might code for putative cis-prenyltransferase responsible for the formation of oligoprenyl/polyprenyl chains.

15) Results of the proteomic analysis should be commented in the context of the paper published by Ponciano G et al., Phytochemistry 2012.

---

## [Author Response]

[…]

Essential revisions:

1) The result from the RTase assay in Figure 7 indicates that the addition of only HRT1 to WRPs results in the incorporation of 14C IPP into products. Does this mean that HRT1 by itself can exert RTase activity on WRPs, or that WRPs still contain unwashed HRBP and REF? I consider that information about WRPs is too limited in the manuscript, just describing, for example, "prepared by incubating 50kRP with 8 mM CHAPS for 1 h at 4^o^C" and "80% of the RTase activity was retained". It will be very helpful for discussion if the authors can indicate the levels of the proteins (and preferably RTase activity) retained on WRPs used for enzyme assay, which would be different from those shown in Figure 1. Moreover, I am not sure if the authors can conclude that the HRT1-HRBP-REF ternary complex is enough for RTase, based on the results from assay using WRPs, which might contain other proteins. The fact that LsCPT3 and LsCPTL2 reconstituted on Hevea WRPs exhibit RTase activity seems insufficient to deny the involvement of the other proteins in the case that such proteins from Hevea can interact also with the lettuce enzymes.

We think that HRT1 itself can exert RTase activity on WRPs, although the WRPs used in the cell-free expression system probably contain some amount of unwashed HRBP and REF, but these amount are considered to be far less than that of recombinantly-expressed HRT1. It is true that the WRPs used in the cell-free expression system were different from that of used for proteomics study. Please see a new supplemental figure (Figure 7—figure supplement 1) describing the difference of protein levels between RP and the RTase activity of WRP that was used for the cell-free expression system (subsection “Reconstitution of RTase on WRPs”). We agree to the reviewer’s comment concerning a possible contribution of other proteins which may interact with the *Hevea* ternary complex or lettuce enzymes. We reshape the main text to mention that a possible involvement of other protein factor(s) and a significance of cPT as a catalytic core for RTase.

2) In the Introduction section, the authors describe "a precise role of NgBR in cPT activity is still unknown owing to a lack of in vitro studies on the function of NgBR family with purified proteins." This is true. In vitro assay with purified proteins was not performed in the previous studies, e.g., ref. 29 about SlCPT3 and SlCPTBP from tomato. But, does the same hold for the present work, which also does not use purified enzyme?

Yes. In the present work, we prepared HRBP which was expressed on WRPs by the cell-free system, and used it as a partially purified enzyme sample. We added the comment describing that the present study does not use purified proteins of NgBR family in the Discussion section, as described in the answer for Comment 1.

3) The explanation in paragraph six of the Discussion section does not make sense. Does HRT2, unlike HRT1, compete with REF on binding to HRBP? The interaction between HRT2 and HRBP looks weak according to the Y2H analysis.

We deleted the description mentioning about interaction affinities between HRT2 and HRBP or HRBP and REF because we are not able to determine the absolute values of interaction affinities in Y2H. Although it is true that the interaction between HRT2 and HRBP looks weak in Y2H analysis, we have detected distinct interaction between HRT2 and HRBP in the plasma membrane in the BiFC assay, indicating translocation of HRBP from the ER to plasma membrane, while REF was sustained on the ER. We added the data in the revised Figure 4.

4) In Figure 7 (left), why was ~40% activity detected in the absence of HRT1 mRNA even though HRT1 was clearly needed for RTase activity?

WRP does not completely lose its RTase activity by the wash treatment with the CHAPS-containing buffer as mentioned in the answer for Comment 1, and the residual activity of WRPs varies depending on batch of 50kRPs, which reflects the background RTase activity of the vector control WRPs. In consequence, relative activity of the control WRPs varies from 20 to 40% of the maximum RTase activity for HRT1-introduced WRPs. It is hard to completely control the residual RTase activity, i.e. residual membrane-bound proteins on RPs, by tuning of washing conditions.

5) Some parts of description are difficult to follow, mainly because the purpose of the experiments are not clearly stated, results are not clearly explained or results are not described in the orderly way.

We corrected the manuscript to state the purpose of the experiments clearly, and to explain the results in the orderly way for better understanding by readers.

6) In many places, RP and detergent-washed Hevea RPs (WRP) are not clearly distinguished. For example in abstract, 'RTase activity enhancement observed for the complex assembled on RPs' in this case RPs must mean WRP. I see others in other places. It is better to be described distinctly. Figure 5—figure supplement 2 is a nice cartoon to follow.

We agree to the comment. We checked the overall manuscript carefully and corrected it to clearly describe RP and WRP distinctly.

7) Figure 1—figure supplement 1 would be better to be shown as a figure instead of supplement as an introduction for the people not familiar with this field.

We agree to the comment. We moved Figure 1—figure supplement 1 to the main part as Figure 1 and renumbered all figures.

8) Formation of the tertiary protein complex HRT1-HRBP-REF has been suggested based on the results of the BiFC and enzymatic assays, unfortunately the in vivo result is missing.

We conducted co-immunoprecipitations from solubilized RP-bound proteins using anti-HRBP and anti-HRT1/HRT2 antibodies. The results suggested the formations of the ternary protein complexes on RPs. We added the results to the manuscript as Figure 6 and descriptions about them in subsection “Formation of the HRT1-HRBP-REF ternary complex on the ER” in the revised manuscript.

9) Classification of proteins applied in the manuscript (Figure 2—source data 1) is unclear. All identified protein subclasses should be shown in this table together with a complete list of proteins. Were any enzymes related to polyisoprenoid biosynthesis (e.g. HMGR, FPPS etc) identified by this method?

We identified a fragment which was annotated as HRT1 or HRT2 as described in the manuscript. There were no enzymes related to polyisopurenoid biosynthesis other than HRT1/HRT2. We added descriptions in the revised manuscript (Results section) and a complete list of proteins to Figure 1—source data 2.

10) No data on the reconstitution of HRT2 in the RP-based in vitro translation system are presented in the manuscript. This fact has to be commented.

In the present study, we tested a reconstitution of HRT2 in the WRP-based in vitro translation system. However, the expression level was very low compared to that of HRT1 for unknown reason, and the resulting WRP with recombinantly expressed HRT2 did not show distinct RTase activity. We added a description of the reconstitution of HRT2 in the Discussion section.

11) The statement concerning the stoichiometry of the HRT1-HRBP complex (1:1) is too speculative. Figure 7 presents only the mRNAs level but not the concentration of the respective proteins. Either western-blot analysis has to be presented or the comment has to be deleted/rephrased.

According to the comment, we deleted the descriptions mentioning the stoichiometry.

12) Authors should commend on the profile of the in vitro obtained products – should a unimodal distribution be expected? A bimodal distribution was observed earlier (Rojruthai et al, 2010).

A bimodal distribution of the rubber products is observed when large rubber particles are mainly used, while small rubber particles (SRPs), correspond to 50kRP in the present study, usually shows a unimodal distribution of the product. This time, we expected a unimodal distribution of the product from in vitro reaction employing SRPs as a platform. We added the comment on the in vitro obtained products in the Results section.

13) What was the efficiency of the in vitro HRT assay? Only relative numbers are presented in Figure 7. It would be interesting to know any absolute data on rubber formation, e.g. the efficiency of IPP incorporation or the amount of rubber formed normalized per amount of RP or protein content.

Using 1 µg of WRPs as platform, when HRT1-HRBP-REF complex was introduced, approximately 0.8 nmol of IPP monomers (which corresponds to approx. 54.4 ng of rubber) were incorporated in the 4 hr reaction of Figure 9. Thus, the efficiency of amount of rubber formed is 13.6 ng of de novo rubber/1 µg of WRPs/1hr enzymatic reaction. We added a description of the rubber amount used in the legends for Figure 9 and Figure 10.

14) Authors have to reshape their concept on the role of additional cis-prenyltransferases, besides HRT1, in NR formation. Moreover, what is the role of HRT2 in this context? The below sentence has to be deleted or rephrased or alternatively additional experiments are required.

Authors neglect (l.258) possible role of polyisoprenoid precursors in rubber biosynthesis and describe HRT1 as a sole cis-prenyltransferase involved in this process.

*'… HRT1 on RP catalyses a de novo rubber formation using allylic primer substrates, GPP, FPP and GGPP, rather than addition of IPP onto the α-terminus of pre-existing rubber molecules in RPs…'*

Interestingly, no rubber formation was observed for the liposome-based system despite the fact that all three proteins HRT1-HRBP-REF were present. In the opinion of this reviewer the Author should provide experimental data to support this statement showing that:

*1) no preexisting polyisoprenoids were present in their RP preparation – what is known about the lipid composition of the monolayer surrounding the RB?*

2) it should be shown that RPs used in the assay do not contain any protein that might code for putative cis-prenyltransferase responsible for the formation of oligoprenyl/polyprenyl chains.

We agree to the comment to reshape our concept on the role of additional cPTs in NR formation. However, as we wrote in above sections (essential revisions 3 and 10), we still don’t know the role and significance of HRT2 in NR biosynthesis because of its unexpected subcellular localizations in *N. benthamiana* cells and its low expression in the cell-free expression system. We reshaped the sentence and add descriptions in the Discussion section so that we do not neglect possible roles of polyisoprenoid precursors and cPTs in rubber biosynthesis. We appreciate the helpful opinions of this reviewer and are now very curious about the lipid composition of WRP.

15) Results of the proteomic analysis should be commented in the context of the paper published by Ponciano G et al., Phytochemistry 2012.

We added a description concerning the proteomic analysis in previous studies.